# Controlling SARS-CoV-2 in schools using repetitive testing strategies

Andrea Torneri[1,2]*, Lander Willem[1,2], Vittoria Colizza[3,4], Cécile Kremer[2], Christelle Meuris[5], Gilles Darcis[5], Niel Hens[1,2]*†, Pieter JK Libin[2,6,7]*†

[1]Centre for Health Economic Research and Modelling Infectious Diseases, University of Antwerp, Antwerp, Belgium; [2]Interuniversity Institute of Biostatistics and statistical Bioinformatics, Data Science Institute, Hasselt University, Hasselt, Belgium; [3]INSERM, Sorbonne Université, Pierre Louis Institute of Epidemiology and Public Health, Paris, France; [4]Tokyo Tech World Research Hub Initiative (WRHI), Tokyo Institute of Technology, Tokyo, Japan; [5]Department of Infectious Diseases, Liège University Hospital, Liège, Belgium; [6]Artificial Intelligence Lab, Department of Computer Science, Vrije Universiteit Brussel, Brussels, Belgium; [7]KU Leuven, Department of Microbiology and Immunology, Rega Institute for Medical Research, University of Leuven, Leuven, Belgium

*For correspondence:
andrea.torneri@uhasselt.be (AT);
niel.hens@uhasselt.be (NH);
pieter.libin@vub.be (PJKL)

†These authors contributed equally to this work

**Abstract** SARS-CoV-2 remains a worldwide emergency. While vaccines have been approved and are widely administered, there is an ongoing debate whether children should be vaccinated or prioritized for vaccination. Therefore, in order to mitigate the spread of more transmissible SARS-CoV-2 variants among children, the use of non-pharmaceutical interventions is still warranted. We investigate the impact of different testing strategies on the SARS-CoV-2 infection dynamics in a primary school environment, using an individual-based modelling approach. Specifically, we consider three testing strategies: (1) *symptomatic isolation*, where we test symptomatic individuals and isolate them when they test positive, (2) *reactive screening*, where a class is screened once one symptomatic individual was identified, and (3) *repetitive screening*, where the school in its entirety is screened on regular time intervals. Through this analysis, we demonstrate that repetitive testing strategies can significantly reduce the attack rate in schools, contrary to a reactive screening or a symptomatic isolation approach. However, when a repetitive testing strategy is in place, more cases will be detected and class and school closures are more easily triggered, leading to a higher number of school days lost per child. While maintaining the epidemic under control with a repetitive testing strategy, we show that absenteeism can be reduced by relaxing class and school closure thresholds.

## Editor's evaluation

This paper evaluates different testing strategies on the SARS-CoV-2 transmission dynamics in a primary school environment and shows that repetitive testing significantly reduces the infection attack rates in the schools. It provides insights into policy design to keep schools open as much as possible in the era of transition from COVID pandemic to endemic.

## Introduction

The SARS-CoV-2 pandemic has caused over 200 million COVID-19 cases and over 4 million deaths around the world up to September 2021 (*World Health Organization, 2022*). Although vaccines have been approved, even for young children, there is an ongoing debate whether such age classes should be vaccinated or prioritized for vaccination (*Wong et al., 2021*). While the contribution of

children in the COVID-19 epidemic is still subject to discussion (*Gaythorpe et al., 2021*), there is a consensus that more infectious variants can cause significant outbreaks among children (*Milne et al., 2022*). Furthermore, recent work indicates that children, who typically undergo an infection with little or no symptoms, might still be highly contagious and as such generate new infections in the community (*Meuris et al., 2021*). As alternative to a vaccination-based strategy, the only means to mitigate outbreaks of SARS-CoV-2 in primary schools, is through non-pharmaceutical interventions, including the use of masks, social distancing, hygienic precautions, and diagnostic testing. Here, the aim of diagnostic testing is to detect and subsequently isolate infected individuals. Therefore, it is important to advance our understanding on how different testing strategies impact primary schools, considering the evolution of SARS-CoV-2 contagiousness through different variants of concerns (VOCs). When defining intervention policies in a school setting, attention needs to be devoted to limiting the number of school days lost. In fact, Engzell et al. evaluated the impact of school closures on students' learning performance, finding that students of age 8–11 years made less progress while learning from home (*Engzell et al., 2021*). Several scientific investigation discussed the use of testing strategies in school settings, for example (*Colosi et al., 2022*; *Leng et al., 2022*; *GOV UK, 2021a*; *GOV UK, 2021b*; *Paltiel and Schwartz, 2021*; *Chang et al., 2020a*; *Hamer et al., 2021*), suggesting that a repetitive testing strategy reduces transmissions in a school context but increases absenteeism. In this work, we explore the effectiveness of testing strategies in a primary school setting by varying factors related to the considerate strategies and school environment, and by testing viral and immunological characteristics representing different SARS-CoV-2 VOCs. To do so, we construct an individual-based model that explicitly represents a set of primary school pupils. These pupils are allocated to a fixed set of classes and are taught by a fixed set of teachers. Through this micro-model, we perform a fine-grained evaluation of testing strategies, keeping track of both the attack rate and the number of school days lost. We conduct experiments considering different $R_0$ values to reflect the increase in infectiousness exhibited by the Delta VoC and we vary the incubation period and the proportion

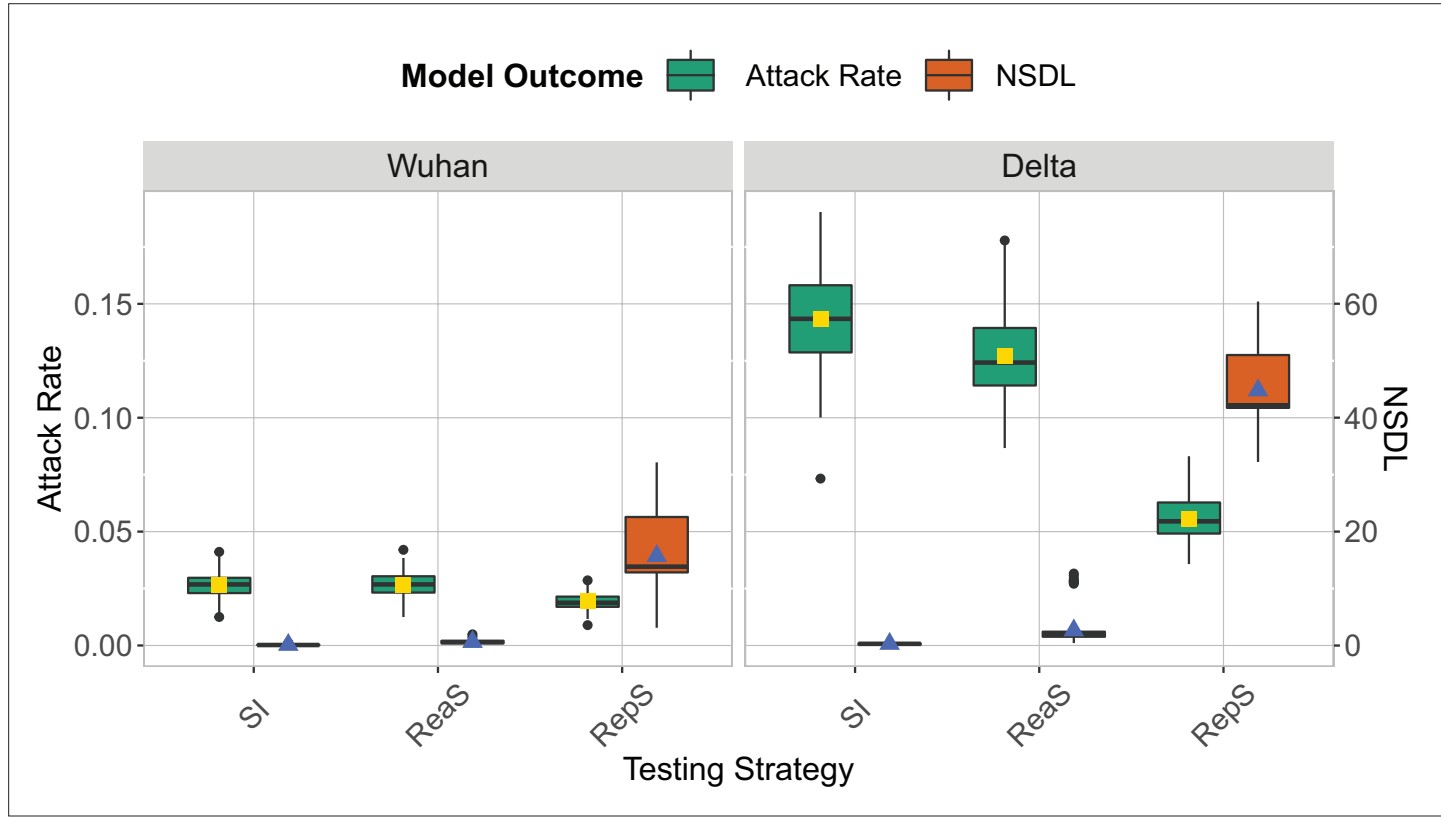

**Figure 1.** We show the base scenario for the Wuhan strain (left panel) and Delta VoC (right panel) for a moderate seeding of 5 seeds per week. In each panel, we consider three testing strategies: symptomatic testing (SI), symptomatic testing in combination with reactive screening (ReaS) and repetitive screening (RepS). For each of the testing strategies we show a boxplot of the attack rate (green boxplot) and NSDL (orange boxplot) together with their mean values (respectively, yellow and blue dots). The epidemic is simulated for 100 days.

of immune individuals to represent the surge of the Omicron VoC. In addition, we investigate the impact of class and school closure thresholds, incubation period, proportion of symptomatic infections, school size and seeding frequency.

## Results

To investigate the efficacy of the testing strategies, we consider both the attack rate (i.e., proportion of the infections generated in the school population, excluding seeded cases) and the average number of school days lost per child (NSDL). In order to differentiate between different phases of the epidemic, we first compare two scenarios that represent the Wuhan strain and the Delta VoC, characterized by a different transmission potential given a contact. Further, we present the case of the Omicron VoC, where we reduce the proportion of immune individuals and we consider a shorter incubation period.

On the one hand, our modelling experiments (details on the simulation model in the Methods section) show (*Figure 1*) that symptomatic isolation results in the infection of a significant proportion of the school population (Wuhan strain, median: 0.03, 95% quantile interval [0.01,0.04]; Delta VoC, median: 0.14, 95% quantile interval [0.10,0.18]). This comes as no surprise, as in our model we assume that 80% of the pupils will go through the infection asymptomatically, and by following this testing policy, we are only able to pick up infections that make up the tip of the iceberg. On the other hand, the attack rate consistently decreases when a testing policy is used that performs a wider screening of the school population, such as reactive testing and repetitive testing. Such policies enable the detection of both symptomatic and asymptomatic cases, that can be subsequently isolated, thereby limiting further transmissions. Among the two screening options, we observe that repetitive approach is the strategy that most reduce the attack rate (Wuhan strain, median: 0.02, 95% quantile interval [0.01,0.03]; Delta VoC, median: 0.05, 95% quantile interval [0.04,0.07]). However, contrary to intuition, our experiments indicate that the reactive screening strategy performs only slightly better than symptomatic isolation (Wuhan strain, median: 0.03, 95% quantile interval [0.01,0.04]; Delta VoC, median: 0.12, 95% quantile interval [0.10,0.17]). This can be explained by the low probability that pupils will be symptomatic when infected, hence a low probability to trigger the reactive screening. When we assume that 80% of infections in children progress asymptomatically, we can expect (by assuming a geometric distribution) that four asymptomatic generations take place, on average, before a symptomatic infection is observed. Therefore, when a reactive screening procedure is triggered by a symptomatic individual, the infected individuals that share a class with this individual might already be recovered or in the end phase of their infectious period. To confirm this reasoning, we simulated a multiple class screening strategy that is triggered when a pupil tests positive. We notice similar attack rates when the screening procedure is repeated (*Appendix 1—figure 4*). Hence, on average, only a limited number of generations can be avoided by employing a reactive screening strategy, when the infection is predominantly driven by asymptomatic carriers. Note that we also assume that only a limited percentage of symptomatic children is detected (30%), due to the fact that many children exhibit only minor symptoms.

To interpret the experimental results with respect to the average number of school days lost per child (NSDL), we need to recognize that children can miss school due to isolation when infected or due to quarantine due to a high risk contact. For the symptomatic isolation strategy, only children with symptoms are isolated, resulting in an average NSDL per child that is directly proportional to the fraction symptomatic cases (Wuhan strain, median: 0.08, 95% quantile interval [0.03,0.3]; Delta VoC, median: 0.3, 95% quantile interval [0.07,0.63]). For reactive screening, additional asymptomatic pupils might be identified, thereby quickly reaching the class or school closure thresholds, with a higher NSDL as a result (Wuhan strain, median: 0.59, 95% quantile interval [0.11,1.34]; Delta VoC, median: 1.99, 95% quantile interval [0.68,11.58]). This effect is most pronounced when we apply repetitive testing, where we effectively detect a high proportion of the infections, thereby rapidly meeting the class and/or school closure thresholds, with a very high NSDL as a consequence (Wuhan strain, median: 13.85, 95% quantile interval [3.50,31.93]; Delta VoC, median: 42.18, 95% quantile interval [32.97,51.64]).

We note that the high NSDL associated with repetitive testing, renders this testing policy impractical. We argue that, by using repetitive testing, more lenient thresholds could be applied, as we are able to detect a larger proportion of cases. We investigate this in *Figure 2* where we remove the school

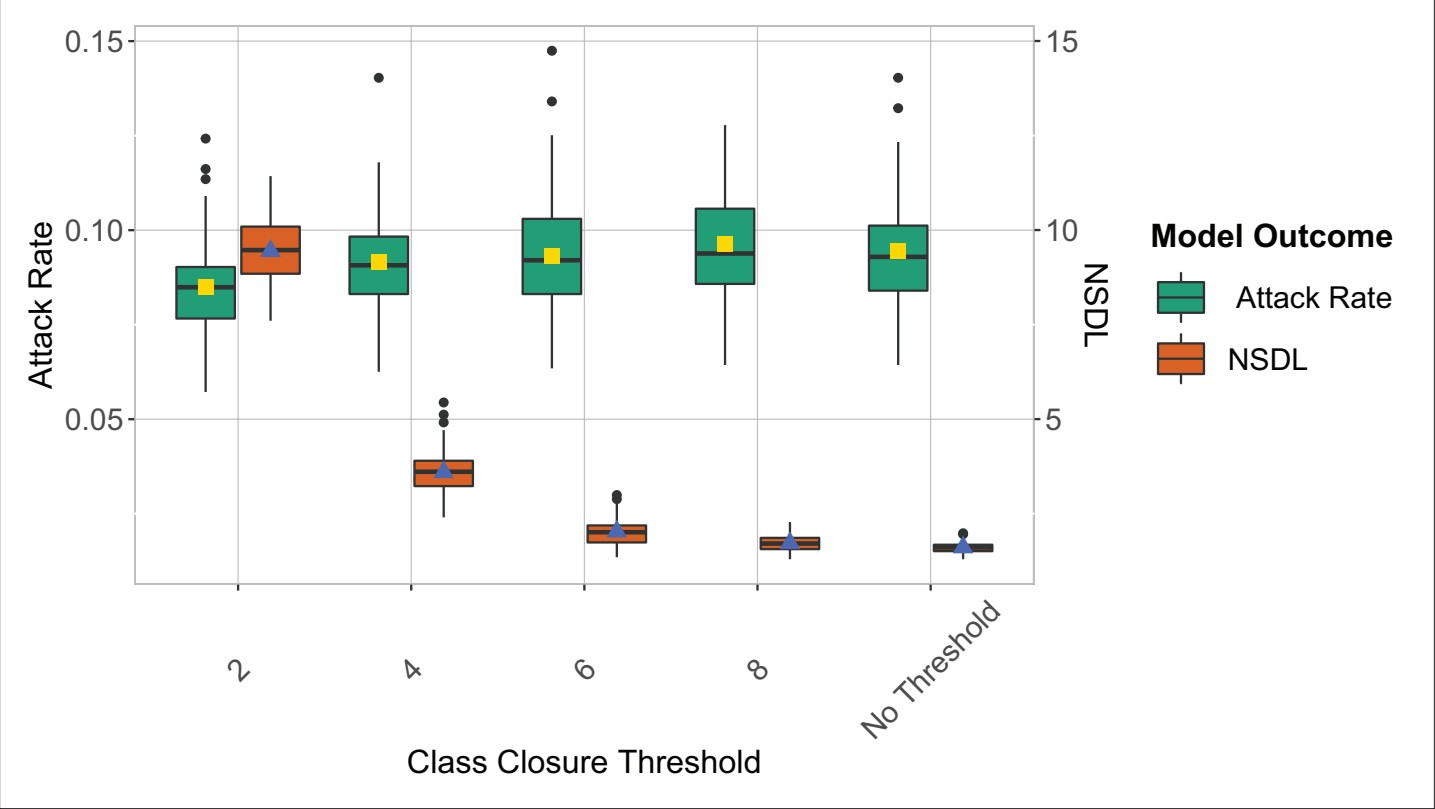

**Figure 2.** We show the repetitive testing strategy in the context of the Delta VoC for a moderate seeding of 5 seeds per week, where we consider different class closure thresholds, and no school closure threshold. The epidemic is simulated for 100 days. This experiment shows that when repetitive testing is in place, a higher class closure threshold has little effect on the attack rate, yet it significantly reduces the NSDL (**Table 1**).

closure threshold, and investigate a set of class closure thresholds while considering a repetitive testing strategy. This experiment confirms that a larger class threshold can be used, with only a limited impact on the attack rate, and that such thresholds result in a more acceptable NSDL. A more stringent school threshold shows a positive effect on controlling the attack rate, but drastically increases the NSDL (**Appendix 1—figure 9**). While the overall trends of the reported measures are similar for the Wuhan and Delta scenarios, the difference between the testing strategies is most pronounced in case of the more infectious virus strain (i.e., the Delta VoC).

In our Omicron scenario with low immunity levels and a shorter incubation period, we observed high attack rates that are more difficult to control (**Appendix 1—figure 20**). To further reduce the number of infections, a twice weekly testing could be considered (**Appendix 1—figure 21**).

In order to challenge some of the assumptions of this study, we conduct a series of sensitivity analyses. We show these results in the Supplementary Information and briefly report the main findings here. We investigate the impact of the amount of weekly introductions, by seeding 1 and 10 cases on a weekly basis, next to the baseline scenario of 5 cases. We note that the impact of additional seeding cases amplifies the attack rate, but overall repetitive testing proved to be robust in regard of this parameter (**Appendix 1—figure 1**). Futher, we notice that a higher attack rate is

**Table 1.** Median and 95% Quantile Interval - Class closure threshold scenario.

| Class Closure Threshold | Summary Measure | Median | 95% Quantile Interval |
|---|---|---|---|
| 2 | Attack Rate | 0.08 | [0.07,0.11] |
| 4 | Attack Rate | 0.09 | [0.07,0.12] |
| 6 | Attack Rate | 0.09 | [0.07,0.12] |
| 8 | Attack Rate | 0.09 | [0.07,0.12] |
| No Threshold | Attack Rate | 0.09 | [0.07,0.11] |
| 2 | NSDL | 9.5 | [7.86, 11.16] |
| 4 | NSDL | 3.6 | [2.50, 4.81] |
| 6 | NSDL | 2.01 | [1.46, 2.83] |
| 8 | NSDL | 1.71 | [1.36, 2.17] |
| No Threshold | NSDL | 1.61 | [1.39, 1.91] |

detected in school with smaller sizes when the seeding number is kept the same (*Appendix 1—figure 17*). When introducing a number of seeds proportional to the school size, the attack rate and NSDL are similar, but a higher stochasticity is observed for the smaller school size (*Appendix 1—figure 18*). Furthermore, the high efficacy of repetitive testing is also observed when varying the level of contact reduction between classes, when considering different levels of immunity in children and adults, in a low prevalence setting and when a high probability of symptomatic infections is considered (*Appendix 1—figures 5–7, 12 and 13*). Next, we assume that asymptomatic individuals are as infectious as symptomatic individuals, as recently observed by *Meuris et al., 2021*. Also in this case, the trends of attack rate and number of school days lost among the different testing strategies are consistent with the baseline scenarios reported above (*Appendix 1—figure 8*). Repetitive testing and reactive screening are less effective when the turnaround time is increased (*Appendix 1—figure 15*). While reactive screening performs similar to symptomatic isolation for a turnaround time of 3 days, repetitive screening is still the strategy that is most successful to reduce the number of infections. In addition, testing strategies show a lower performance when a shorter incubation period is considered (*Appendix 1—figure 14*). We also consider a repetitive testing scenario where we test the entire school population twice per week, which shows that such a strategy squashes a highly contagious epidemic such as driven by the Delta VoC (*Appendix 1—figures 2 and 3*) and reduces the attack rate of an immune-evasive VoC with shorter incubation period, as the Omicron VoC (*Appendix 1—figure 21*). In addition, we noticed a decrease in the NSDL when assumptions on school and class closure are relaxed for the Delta VoC (*Appendix 1—figure 3*) compared to a single repetitive testing strategy. However, when school and class thresholds are present and the Delta Voc considered (*Appendix 1—figure 2*), or when the thresholds are relaxed and the Omicron VoC assumed (*Appendix 1—figure 21*), the NSDL increases if testing twice per week. Considering a repetitive testing strategy, we also tested the compliance to testing, showing that attack rate decreases and NSDL increases when compliance is increased (*Appendix 1—figure 16*). Interestingly, in our experimental setting, a compliance of 60% leads to a similar attack rate than a compliance of 100%.

## Discussion

This simulation study compares the efficacy of testing strategies for mitigating COVID-19 outbreaks in a school setting. We evaluated such strategies computing both the attack rate and the number of school days lost. The former quantity is related to the risks of importations into households and communities, and of complications from infection, e.g. long COVID and Multysystem Inflammatory Syndrome, while the latter to educational disruption.

Throughout all simulated scenarios, a repetitive testing procedure is shown to be most efficient to reduce the attack rate. Simulations indicate that such a testing strategy limits the number of transmission events even when no class and school closures are in place. The low efficacy of the symptomatic testing and reactive screening procedures is explained by the asymptomatic nature of SARS-CoV-2 infections, especially for children. In fact, when surveillance is based just on the onset of symptoms, asymptomatic carriers avoid detection and intervention, sustaining the spread of the virus.

Class and school closures affect the number of school days lost of healthy children. To limit the learning loss caused by such closures, a control strategy in which only infected cases are isolated would be optimal. This is the aim of the a repetitive testing strategy for which no school or class thresholds are considered. Within our experimental settings, we observe that repetitive testing can keep transmission under control and limit the number of school days lost.

In our experiments, we consider PCR tests as gold standard, as we argue that the available testing infrastructure is most appropriate for performing reactive and repetitive screening procedures. To make this procedure more efficient, a class pooling approach could be used to reduce the number of samples to be analyzed (*Libin et al., 2021*). To further reduce the number of required PCR tests, the use of a repetitive testing strategy can be targeted to areas where prevalence is particularly high.

The viral input parameters chosen in the simulation study were set to describe the spreading of COVID-19. However, other infectious diseases can easily be represented by incorporating the specific transmission characteristics of the respective pathogens in the simulator. Especially in the case of emerging epidemics or pandemics with higher contagiousness in child-to-child interactions and/or a higher severity for children, an appropriate testing strategy in a school setting is pivotal to dampen

epidemic spread. By using the simulation model presented in this paper, ad-hoc testing strategies can be easily simulated offering valuable insights in controlling epidemics.

We assume that teachers are allocated to specific classes and are assumed to interact only with individuals with whom they share the same class. This means that the interaction between teachers in the school environment is limited. We argue that this is a reasonable assumption at this stage of the epidemic, where a large proportion of teachers is either immune or vaccinated. In order to add such functionality to the model, an additional contact structure could be added to the model in which teachers meet, that is, a teacher room, to be informed by the contact frequencies between adults in a school environment (*Verelst et al., 2021*).

In the baseline scenario, we assume perfect compliance by school individuals for both the reactive and repetitive screening. We argue that this is a reasonable assumption, as the threshold for participating in saliva sampling is low, and societal awareness and support for this policies can be achieved, via prompt governmental communication. Nonetheless, we investigate the effect of compliance to testing for the repetitive testing strategy. Interestingly, in our experimental setting a similar attack rate is observed for a compliance level of 60% and 100%.

## Methods
### Individual-based primary school model
We construct an individual-based model to describe COVID-19 outbreaks in a primary school setting, which we briefly introduce in this section We refer to the Supplementary Information for a full description of the model. Children are assigned to classes and we simulate interactions among children both within and between classes. Teachers are allocated to specific classes and are assumed to interact only with individuals with whom they share the same class. We assume that symptomatic individuals develop symptoms at the peak of their infectiousness, at which they can be detected and placed in isolation for 10 days. We implement three testing policies aimed at mitigating school outbreaks:

- *Symptomatic Isolation* (SI). Symptomatic individuals are detected with probability $p_D$ and tested. Individuals that test positive are put in isolation.
- *Reactive Screening* (ReaS). Symptomatic individuals are detected with probability $p_D$ and tested. Individuals that test positive are put in isolation. In addition, all members of the class where this case originates from are also tested. When any additional cases are detected, these individuals are also put in isolation.
- *Repetitive Screening* (RepS). All of the school's members are tested on a repetitive basis once per week. All individuals that test positive are put in isolation.

All testing policies will close a class when the number of infections in this class exceeds two cases. Analogously, all testing policies will close the school when the number of infections over all classes exceeds 20 cases. When the class, or school, threshold is triggered, the respective class, or the entire school, is closed for 10 days. The length of isolation and of class/school closure is set according to viral clearance observations (*Chang et al., 2020b*), and in line with isolation policies in place in European countries in the first half of 2021. Infection counts are recorded in a 14-day time window to determine class and school closures. We assumed a weekly screening as the baseline scenario because a strategy based on a single test can be more easily applied at a national level when a high amount of tests need to be quickly analyzed. However, we also consider a repetitive screening strategy based on twice weekly testing. The assumptions on class and school thresholds, and on the frequency of weekly testing are challenged in a sensitivity analysis, which we discuss in the Results section.

### Experimental framework
Model parameters are set to describe COVID-19 spreading. We represent both the Wuhan strain of SARS-CoV-2 and the Delta variant by setting different transmission potentials given a contact, informing such values from the literature (*Li et al., 2020*; *Burki, 2021*). We consider a distinct detection probability of symptoms $p_D$ for children ($p_D = 0.3$) and adults ($p_D = 0.5$), as children typically exhibit mild symptoms that are easily overlooked (*Sinha et al., 2020*). Children are set to be halve as susceptible as adults (*Davies et al., 2020*). We assume that 30% of school children are immune due to prior infection, and that 90% of the teachers are immune, due to their vaccination status or due to prior infection (*Sciensano, 2022*).

The simulated testing procedure accounts for the use of PCR tests on saliva or throat washing samples. The sensitivity of such tests is set to 86%, and there is a one day delay in reporting the result (*Butler-Laporte et al., 2021*). Recent reports show that the performance of saliva sampling in combination with PCR testing is on par with nasopharyngeal swab sampling in combination with PCR testing (*Wyllie et al., 2020*). We assume full compliance to testing, that could potentially be reached since saliva sampling is less invasive compared to other specimen collection procedures. Infectious individuals become PCR detectable 2 days after infection, as previously assumed (*Torneri et al., 2020*; *Torneri et al., 2021*). For the reactive screening testing policy, we assume that there is a one day screening delay.

Every simulated week, five susceptible children are assumed to acquire infection outside the school environment, accounting for disease importation or seeding. The epidemic is simulated for 100 days and we consider an ensemble of 100 simulation runs to present our final results. The number of simulations was selected allowing for producing clear and stable results, and we show the full distribution of the different statistics, such that the reader can directly interpret the full scope of the simulation results. For each simulated outbreak, we compute two summary measures that account, for the number of transmissions at school and absenteeism, respectively. The former is defined as the total number of cases (minus the index cases) divided by the the number of pupils in the school, and we refer to this quantity as the attack rate. The latter is defined as the sum of the school day lost divided by the school size, and we refer to this quantity as number of school days lost (NSDL).

## Acknowledgements

LW and PJKL acknowledge support from the Research Foundation Flanders (FWO, fwo.be) (post-doctoral fellowships 1234620 N and 1242021 N). NH and PJKL acknowledge support from the European Research Council (ERC) under the European Union's Horizon 2020 research and innovation programme (grant number 101003688—EpiPose project). NH and AT received funding from the European Research Council (ERC) under the European Union's Horizon 2020 research and innovation programme (grant number 682540—TransMID project). NH acknowledge funding from the Antwerp Study Centre for Infectious Diseases (ASCID) and the chair in evidence-based vaccinology at the Methusalem-Centre of Excellence consortium VAX-IDEA. CM received funding from the "Fondation Léon Fredericq" and the "Fond d'investissement de recherche scientifique" from the CHU of Liège. GD received "Post-doctorate Clinical Master Specialists" funding from the Fund for Scientific Research (F.R.S.–FNRS, frs-fnrs.be). We used computational resources and services provided by the Flemish Supercomputer Centre (VSC), funded by the FWO and the Flemish Government. This project was supported by the VERDI project (101045989), funded by the European Union. Views and opinions expressed are however those of the author(s) only and do not necessarily reflect those of the European Union or the Health and Digital Executive Agency. Neither the European Union nor the granting authority can be held responsible for them.The funding agencies had no role in study design, data collection and analysis, decision to publish, or preparation of the manuscript.

## Additional information

### Competing interests

Niel Hens: Reviewing editor, eLife. The other authors declare that no competing interests exist.

### Funding

| Funder | Grant reference number | Author |
| --- | --- | --- |
| Fonds Wetenschappelijk Onderzoek | 234620N | Lander Willem |
| Fonds Wetenschappelijk Onderzoek | 1242021N | Pieter JK Libin |

| Funder | Grant reference number | Author |
|---|---|---|
| Horizon 2020 - Research and Innovation Framework Programme | 101003688 | Niel Hens |
| European Research Council | 68254 | Niel Hens |
| Antwerp Study Centre for Infectious Diseases (ASCID) | | Niel Hens |
| CHU Liege | | Christelle Meuris |
| F.R.S. - FNRS | | Gilles Darcis |
| European Union | VERDI project (101045989) | Niel Hens Andrea Torneri |

The funders had no role in study design, data collection and interpretation, or the decision to submit the work for publication.

## Author contributions

Andrea Torneri, Conceptualization, Data curation, Formal analysis, Investigation, Methodology, Software, Writing – original draft, Writing – review and editing; Lander Willem, Investigation, Methodology, Software, Writing – original draft, Writing – review and editing; Vittoria Colizza, Cécile Kremer, Christelle Meuris, Gilles Darcis, Conceptualization, Methodology; Niel Hens, Conceptualization, Funding acquisition, Investigation, Methodology, Project administration, Supervision, Writing – original draft, Writing – review and editing; Pieter JK Libin, Conceptualization, Data curation, Formal analysis, Investigation, Methodology, Project administration, Software, Supervision, Visualization, Writing – original draft, Writing – review and editing

## Author ORCIDs

Andrea Torneri http://orcid.org/0000-0002-4322-0770
Vittoria Colizza http://orcid.org/0000-0002-2113-2374
Niel Hens http://orcid.org/0000-0003-1881-0637
Pieter JK Libin http://orcid.org/0000-0003-3906-758X

## Decision letter and Author response

Decision letter https://doi.org/10.7554/eLife.75593.sa1
Author response https://doi.org/10.7554/eLife.75593.sa2

# Additional files

## Supplementary files

• Transparent reporting form

## Data availability

The current manuscript is a computational study, so no data have been generated for this manuscript. Source code of the individual-based model was implemented in R (version: R/3.6.0-foss- 2018a- bare) and is freely available in a Zenodo repository at the following DOI: 10.5281/zenodo.6488473. Modelling code is also uploaded as source code on a publicly available Github repository (https://github.com/AndreaTorneri/TestingStrategies, copy archived at swh:1:rev:5b6845b34f9d9a98d7f9438c2b9ffdac00db0a6b).

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

## Appendix 1

### Supplementary information

### Individual-based school model

#### Transmission model

We extend the SARS-CoV-2 transmission model presented by *Torneri et al., 2021* to investigate school settings. In this model, the infection dynamic is described as follows. Individuals are initially susceptible and once infected, they enter the exposed stage. The infection can be asymptomatic or symptomatic. Symptomatic individuals develop symptoms after a pre-symptomatic period. Each symptomatic individual is assumed to show symptoms at the peak of their infectiousness, as indicated by literature findings (*Sun et al., 2021*; *He et al., 2020*). After infection, individuals will eventually recover, after which they are assumed to be immune to reinfection.

Infection events are simulated with a counting process approach. First, contacts between individuals are generated. Contacts are effectives, i.e., lead to the transmission of the virus, according to a Bernoulli trial, based on the time since infection. Effective contacts that take place between susceptible and infectious individuals result in infection events. The probability that a contact is effective is composed of two factors: the infectivity measure, $\nu(t)$ and the transmission potential $q$, which accounts for the transmissibility of the pathogen and the susceptibility of the exposed individual. In this context, the basic reproduction number of an infectious disease is approximated with the mean number of effective contacts infectious individuals generates in a fully susceptible population throughout their infectious periods (*Torneri et al., 2021*).

The infectivity measure $\nu(t)$ is defined over the exposed and infectious period of the infected individual and is set to represent the shape of the viral load curve for a SARS-CoV-2 infection, under the assumption that a higher amount of virus corresponds to a higher transmission probability (*Buonanno et al., 2020*). Based on literature findings, we define an infectivity measure that peaks at symptom onset and lasts 10 days, on average (*Zhou et al., 2020b*; *Zhou et al., 2020a*; *Kim et al., 2020*; *Long et al., 2020*; *Liu et al., 2020*; *He et al., 2020*; *Cevik et al., 2021*). In addition, $\nu(t)$ has an initial plateau with value zero that accounts for the exposed phase.

Asymptomatic and symptomatic individuals are assumed to have the same viral progression, as argued in *Zou et al., 2020*; *Zhou et al., 2020a*, but we introduce a different level of infectiousness between infectious individuals based on the clinical outcome. Precisely, the relative infectiousness of asymptomatic compared to symptomatic is 0.5 (*Davies et al., 2020*).

### School classes, pupils, and teachers

We consider a population of children in primary school (6–12 years old), where each child is randomly allocated to a class. To this end, we construct a set of classes of which the size is sampled from a probability mass function informed by a survey on Belgian primary schools (https://www.agodi.be/nieuwe-omkadering-basisonderwijs), up until at least 1000 pupils are allocated to these classes.

In the school, we consider teaching and supportive staff, to which we will refer as teachers from this point forward for brevity. The number of teachers is proportional to the number of pupils (ratio $\frac{1}{9}$) (https://www.vlaanderen.be/publicaties/vlaams-onderwijs-in-cijfers). We consider different contact ratios within $\lambda_w$ and between $\lambda_b$ classes, and assume that teachers have contacts only with children and other teachers of their allocated classes. Instead, children can have contact with children of the same class and children of different classes. The within and between classes contact rates for children are set accordingly to a contact data survey that took place in Belgium (*Hoang et al., 2019*). The within class contact rate is given by number of contacts that take place in primary schools and last more than 1 hr ($\lambda_w = 6.62$). The between class contact rate is computed as the the number of contacts that last less than 1 hr ($\lambda_b = 2.5$). However, we assumed that in a pandemic setting the number of between class contacts is reduced. In the baseline scenario, we assumed that the between contact rate in a COVID-19 pandemic scenario is 30% of $\lambda_b$. We test such assumption in the sensitivity analysis by varying this proportion among 20%, 50% and 90%.

## Sensitivity analysis
### Seeding number

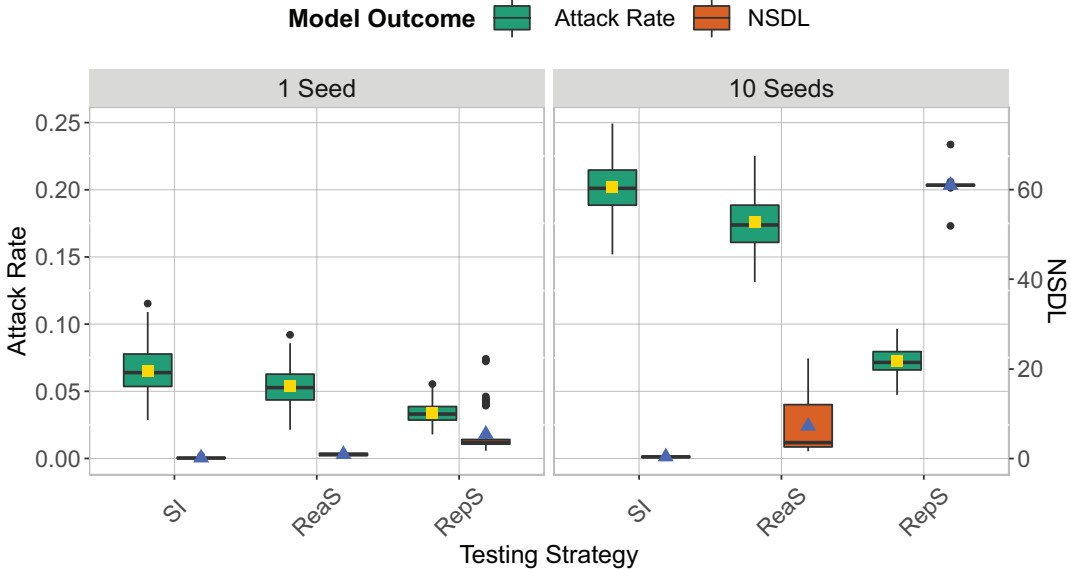

**Appendix 1—figure 1.** We compare the testing strategies in the context of the Delta VoC for a seeding of 1 seed per week (left panel) and 10 seeds per week (right panel). School and class thresholds are set, respectively, to 20 and 2 detected cases. The epidemic is simulated for 100 days. This experiment shows that increasing the number of seeds leads to an increase in both the attack rate and NSDL.

### Number of tests per week

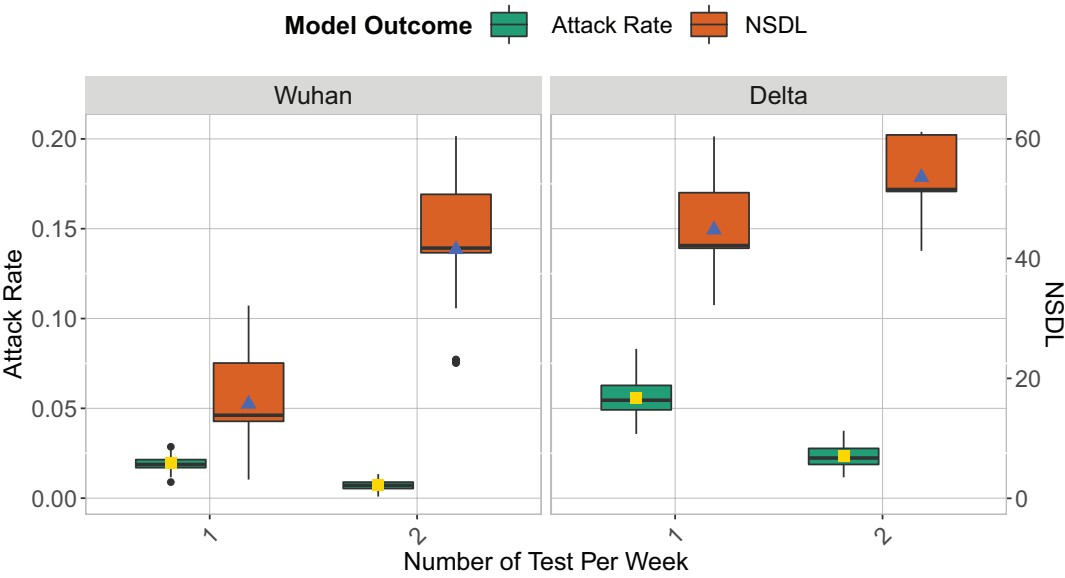

**Appendix 1—figure 2.** We show the repetitive testing strategy in the context of the Wuhan strain and the Delta VoC for a moderate seeding of 5 seeds per week, where we consider a repetitive testing strategy for which the entire school population is tested either once or twice per week. We consider class closure threshold of 2 and school closure threshold of 20 cases. The epidemic is simulated for 100 days. This experiment demonstrates that for a highly infectious virus strain, repetitive testing can further reduce the number of transmissions at school while increasing the NSDL.

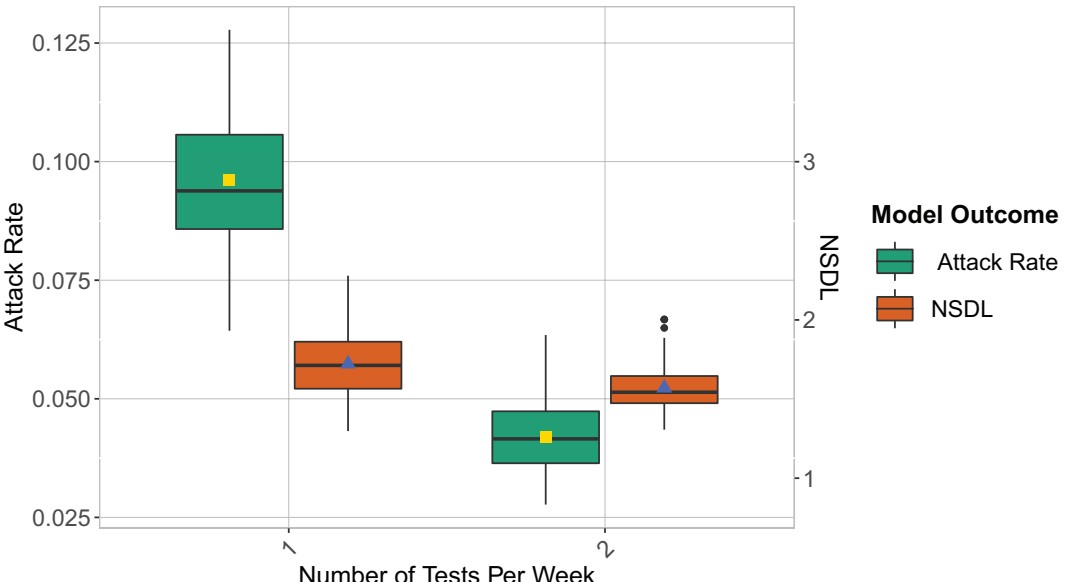

**Appendix 1—figure 3.** We show the repetitive testing strategy in the context of the Wuhan strain and the Delta VoC for a moderate seeding of 5 seeds per week, where we consider a repetitive testing strategy for which the entire school population is tested either once or twice per week. We consider class closure threshold of 8 and no school closure threshold. The epidemic is simulated for 100 days. This experiment demonstrates that twice testing can reduce the number of transmissions at school and the NSDL when assumptions on class and school thresholds are relaxed.

## Multiple screenings

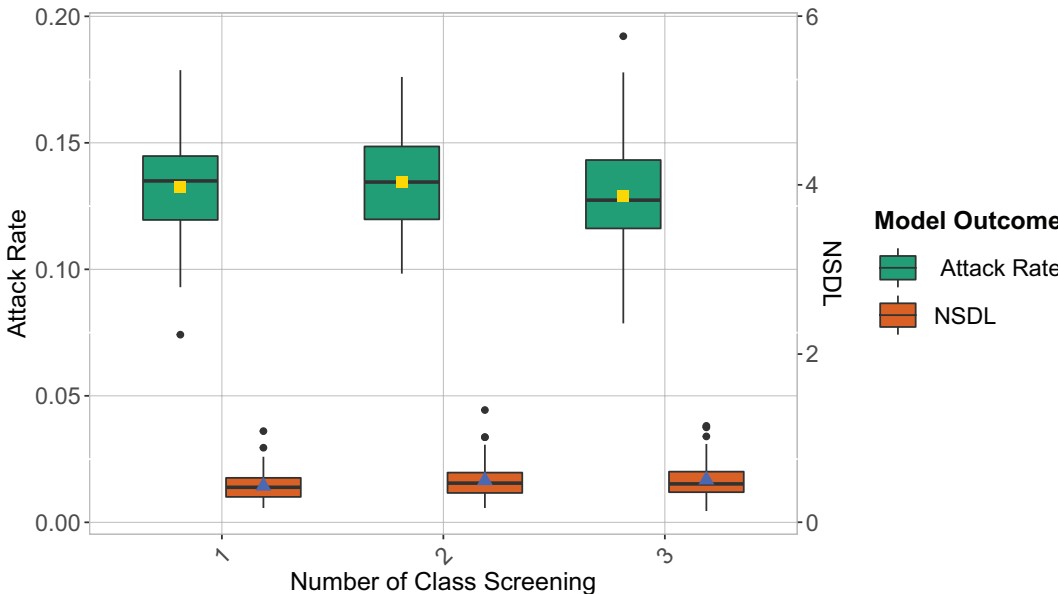

**Appendix 1—figure 4.** We compare the reactive screening strategy in the context of the Delta VoC for a moderate seeding of 5 seeds per week, when varying the number of screening. The class closure threshold is of eight detected cases, and there is no school closure threshold. The epidemic is simulated for 100 days. This experiments shows that increasing the number of screening has a small effect on the attack rate and NSDL.

## Between-classes contact rate

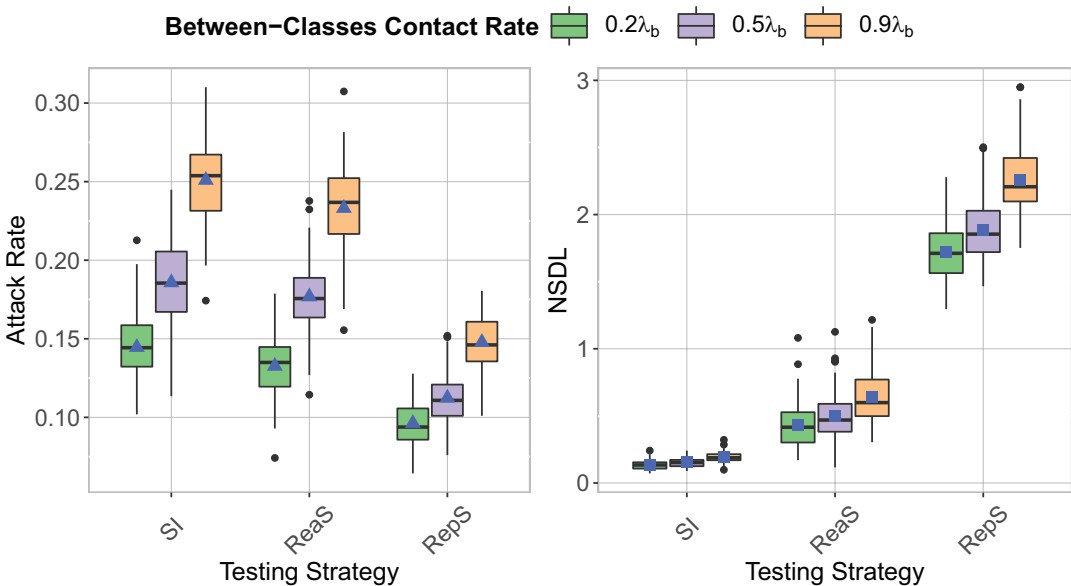

**Appendix 1—figure 5.** We compare testing strategy in the context of the Delta VoC for a moderate seeding of 5 seeds per week, when varying the proportion of between classes contacts compared to a pre-pandemic scenario. The class closure threshold is of eight detected cases, and there is no school closure threshold. The epidemic is simulated for 100 days. This experiments shows that increasing between classes contact rate increases both the attack rate (left panel) and the NSDL (right panel).

## Immune population proportion

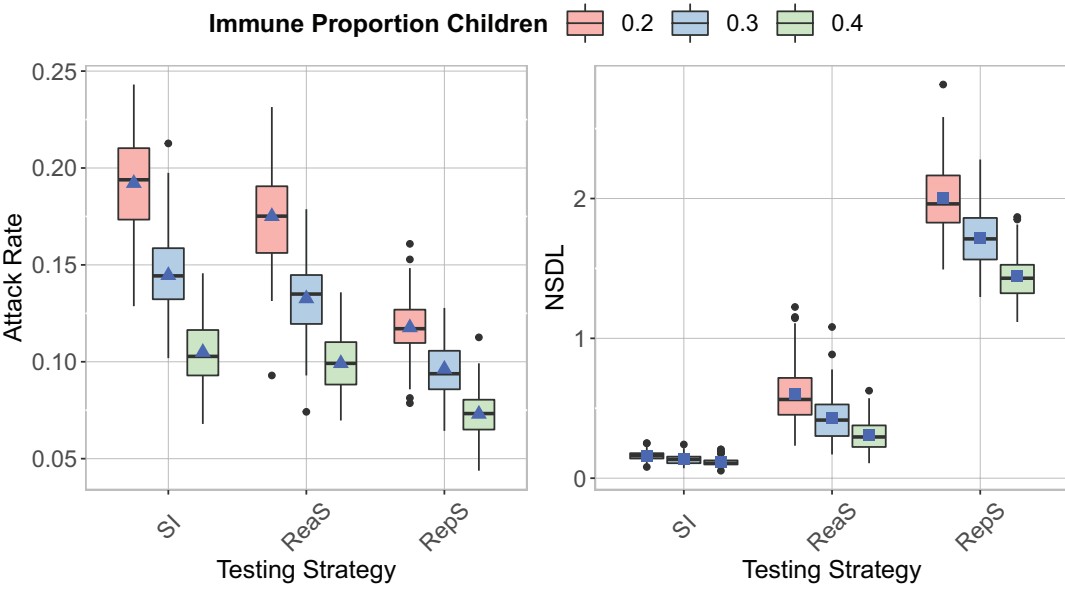

**Appendix 1—figure 6.** We compare the testing strategies in the context of the Delta VoC for a moderate seeding of 5 seeds per week, when varying the proportion of immune children. The class closure threshold is of eight detected cases, and there is no school closure threshold. The epidemic is simulated for 100 days. This experiments shows that increasing the proportion of immune children decrease both the attack rate (left panel) and the NSDL (right panel) for all the testing strategies.

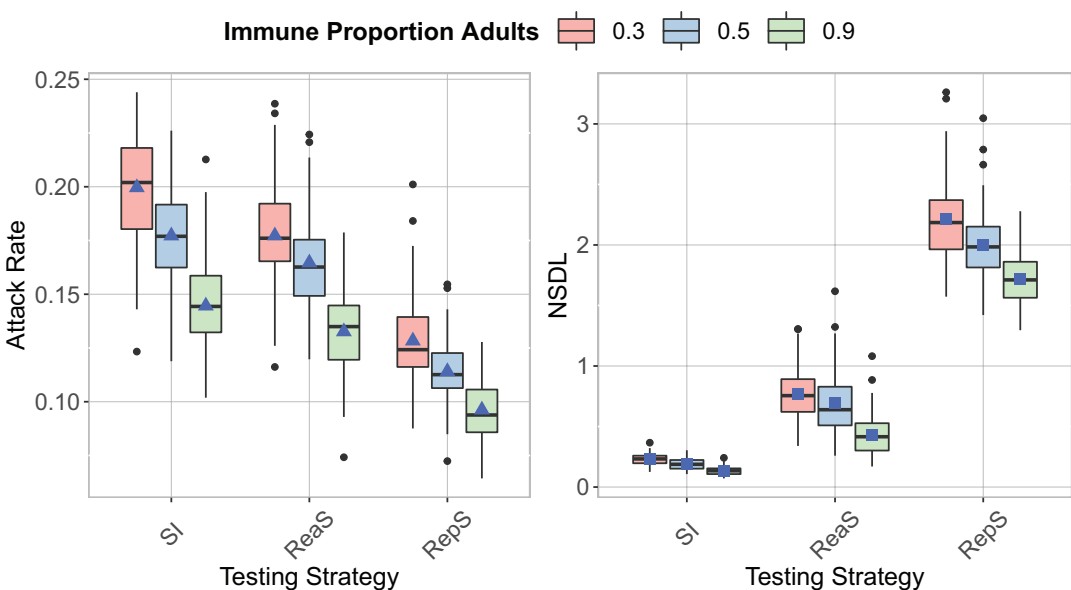

**Appendix 1—figure 7.** We compare the testing strategies in the context of the Delta VoC for a moderate seeding of 5 seeds per week, when varying the proportion of immune adults. The class closure threshold is of eight detected cases, and there is no school closure threshold. The epidemic is simulated for 100 days. This experiments shows that increasing the proportion of immune adults decrease both the attack rate (left panel) and the NSDL (right panel) for all the testing strategies.

## Increased infectivity asymptomatic carriers

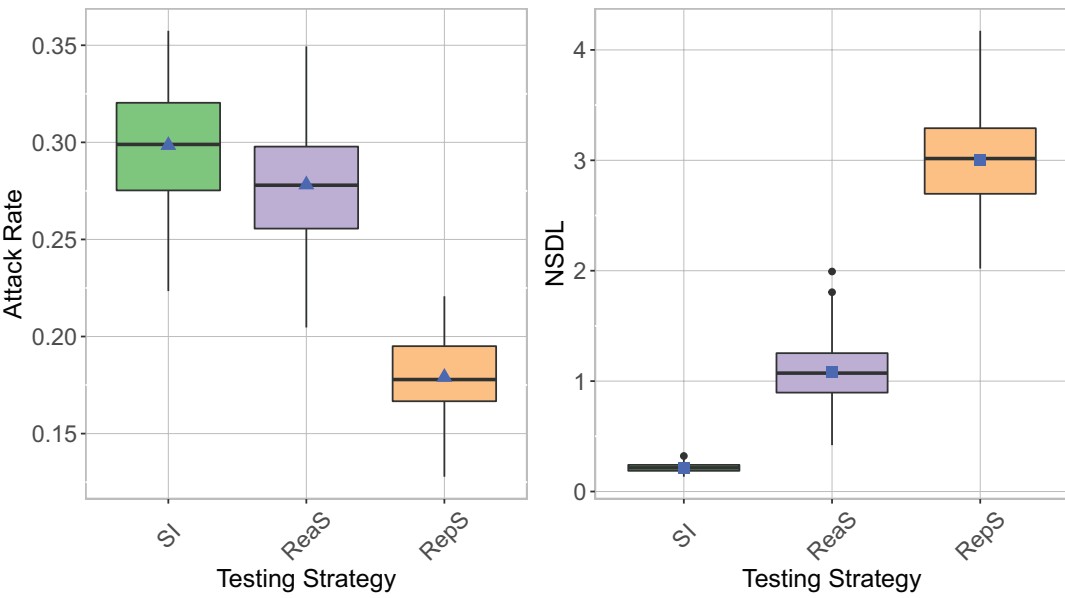

**Appendix 1—figure 8.** We show the testing strategies in the context of the Delta VoC for a moderate seeding of 5 seeds per week, where we consider asymptomatic carriers to be as infectious as the symptomatic ones. The class closure threshold is of eight detected cases, and there is no school closure threshold. The epidemic is simulated for 100 days. The repetitive screening strategy is shown to reduce the attack rate (left panel), while leading to a higher NSDL (right panel).

School closure threshold

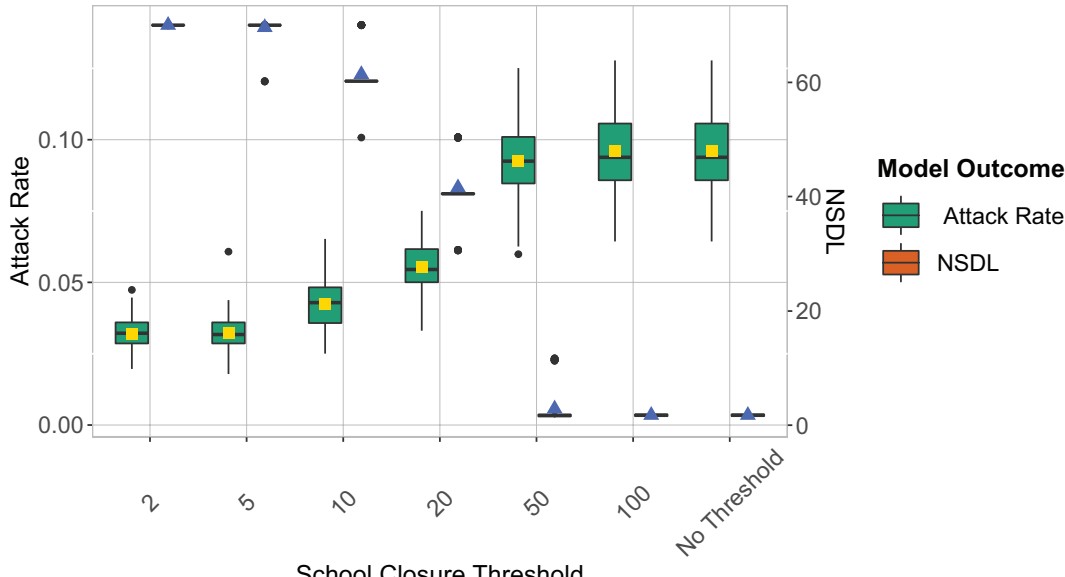

**Appendix 1—figure 9.** We show the repetitive testing strategy in the context of the Delta VoC for a moderate seeding of 5 seeds per week, where we consider different school closure thresholds. The class closure threshold is set to eight detected cases. The epidemic is simulated for 100 days. This experiments shows that a low school closure threshold decreases the attack rate but it increases the NSDL.

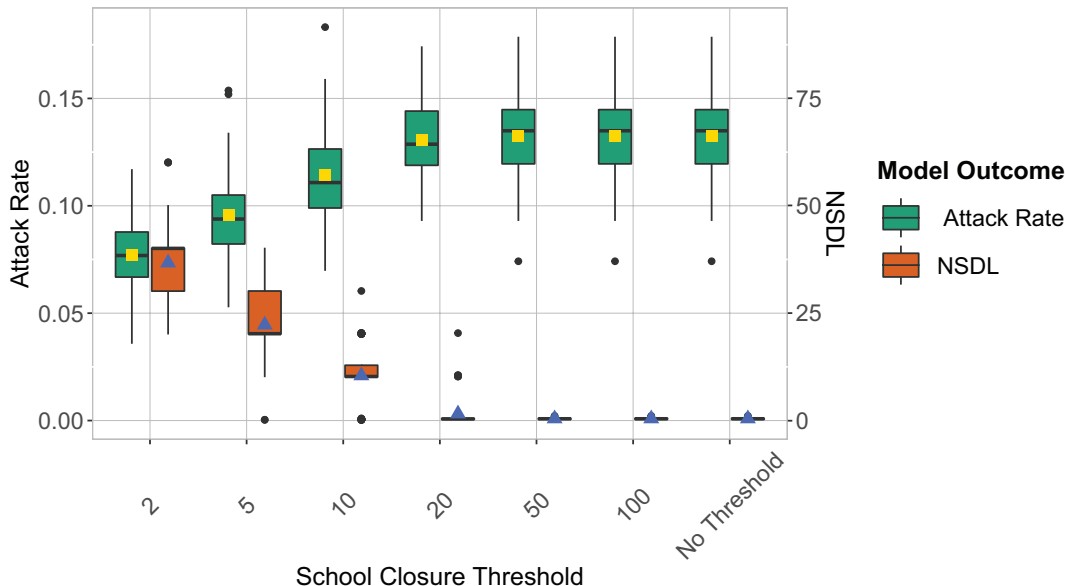

**Appendix 1—figure 10.** We show the reactive screening strategy in the context of the Delta VoC for a moderate seeding of 5 seeds per week, where we consider different school closure thresholds. The class closure threshold is set to eight detected cases. The epidemic is simulated for 100 days. This experiments shows that a low school closure threshold decreases the attack rate but it increases the NSDL.

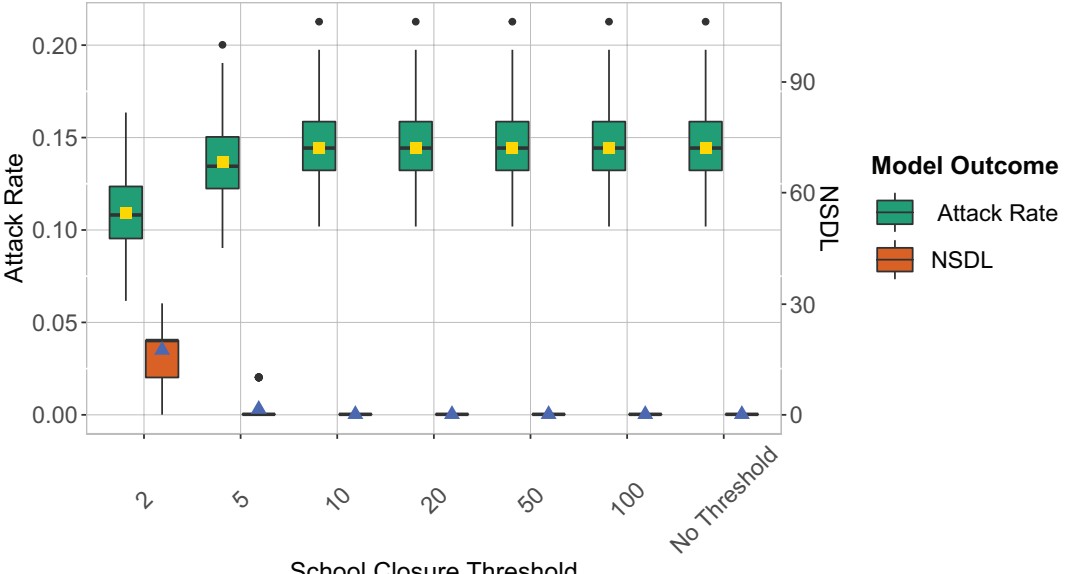

**Appendix 1—figure 11.** We show the symptomatic isolation strategy in the context of the Delta VoC for a moderate seeding of 5 seeds per week, where we consider different school closure thresholds. The class closure threshold is set to 8 detected cases. The epidemic is simulated for 100 days. This experiments shows that a low school closure threshold decreases the attack rate but it increases the NSDL.

## Low seeding scenario

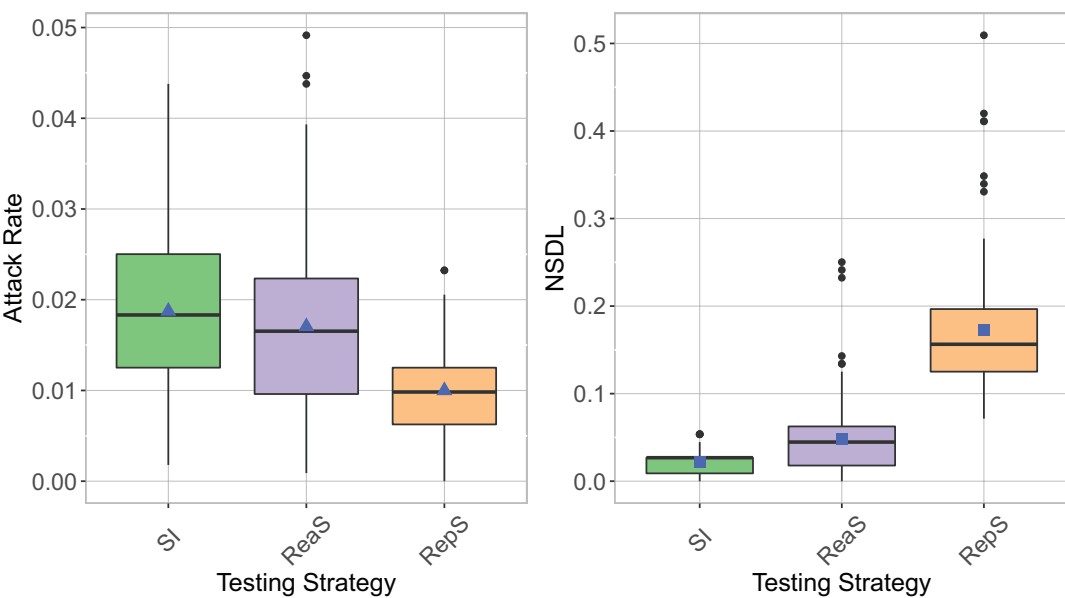

**Appendix 1—figure 12.** We show the testing strategies in the context of the Delta VoC for a seeding of 1 seeds per month, where we consider different school closure thresholds. The class closure threshold is set to eight detected cases. The epidemic is simulated for 100 days. The repetitive screening strategy is shown to decrease the attack rate (left panel) compared to reactive screening and symptomatic isolation, while increasing the NSDL (right panel).

## Probability of symptomatic infections

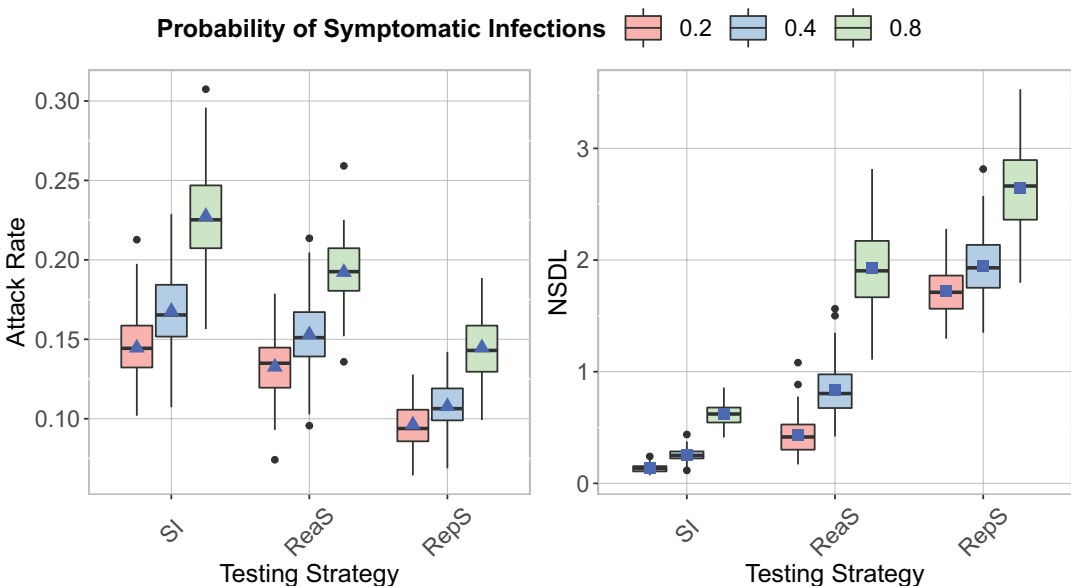

**Appendix 1—figure 13.** We show the testing strategies in the context of the Delta VoC for a moderate seeding of 5 seeds per week, where we consider different probability of having a symptomatic infection. The class closure threshold is set to 8 detected cases and there is no school closure threshold. The epidemic is simulated for 100 days. This experiment shows that to an increase in the probability of having a symptomatic infection it corresponds an increase in the attack rate (left panel) and the NSDL (right panel).

## Incubation period

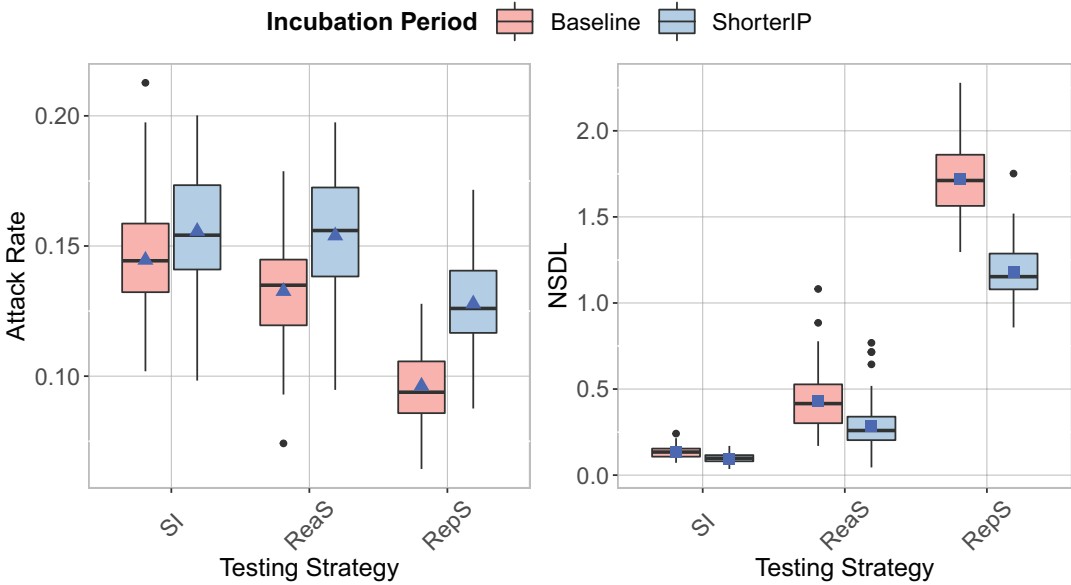

**Appendix 1—figure 14.** We show the testing strategies in the context of the Delta VoC for a moderate seeding of 5 seeds per week, where we consider different incubation period. The class closure threshold is set to eight detected cases and there is no school closure threshold. The epidemic is simulated for 100 days. The repetitive screening strategy is shown to decrease the attack rate (left panel) compared to reactive screening and symptomatic isolation, while increasing the NSDL (left panel). Reactive screening and repetitive screening are less effective in reducing the attack rate when the incubation period is shorter.

Test result delay

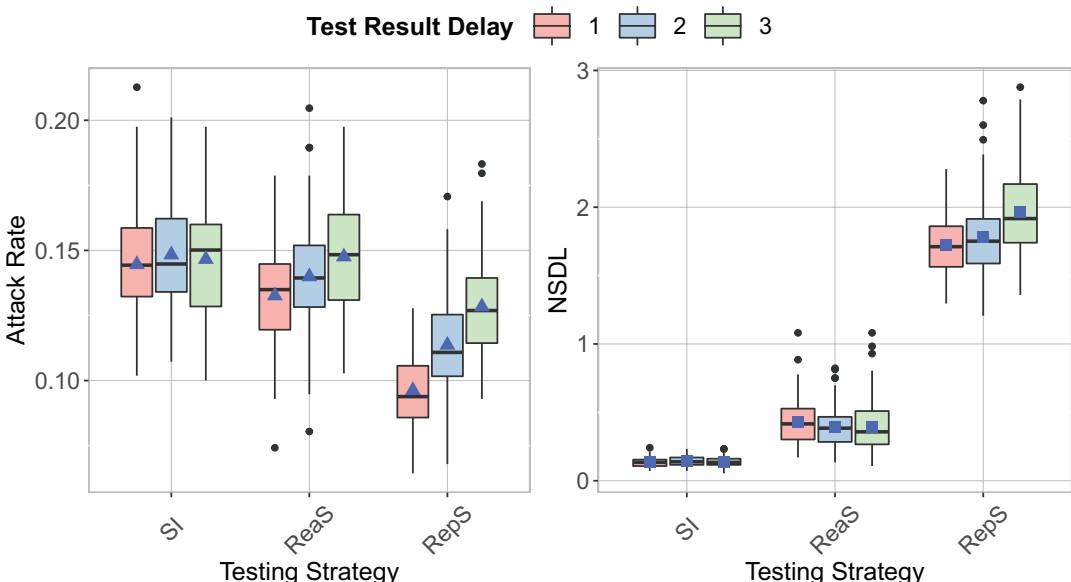

**Appendix 1—figure 15.** We show the testing strategies in the context of the Delta VoC for a moderate seeding of 5 seeds per week, where we consider different turnaround time for the test result. The class closure threshold is set to 8 detected cases and there is no school closure threshold. The epidemic is simulated for 100 days. An increase in the attack rate is shown for repetitive testing and reactive screening with an increase in the turnaround time (left panel). The NSDL increases with an increase in the turnaround time, while a similar trend is shown for the other strategies (right panel).

Compliance

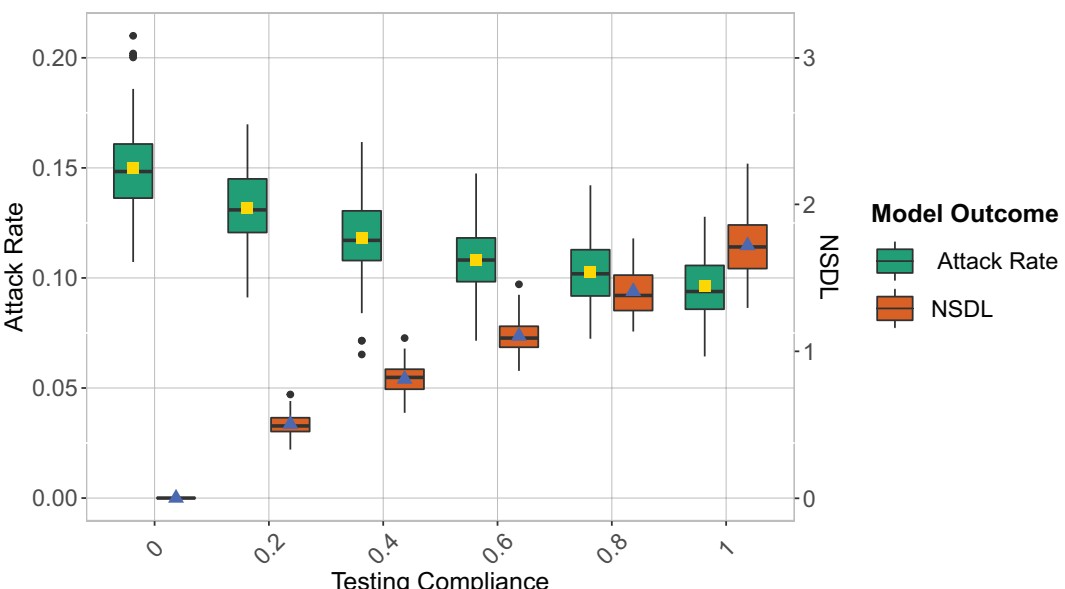

**Appendix 1—figure 16.** We show the repetitive testing strategies in the context of the Delta VoC for a moderate seeding of 5 seeds per week, when varying the compliance to testing. The class closure threshold is set to eight detected cases and there is no school closure threshold. The epidemic is simulated for 100 days. For an increase in the compliance the attack rate decreases and the NSDL increases.

School size

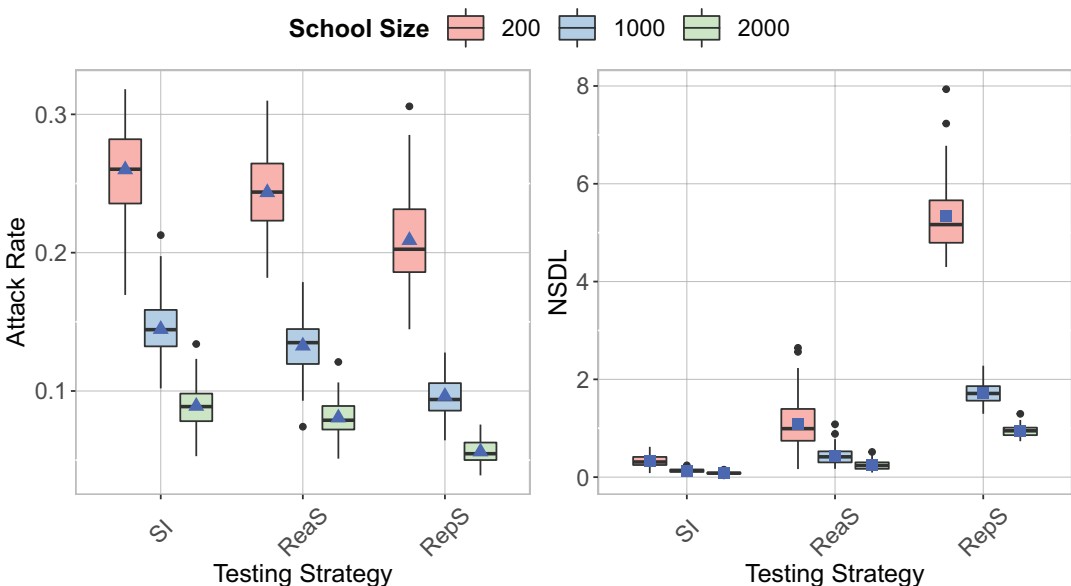

**Appendix 1—figure 17.** We show the testing strategies in the context of the Delta VoC for a moderate seeding of 5 seeds per week, when varying the school size. The class closure threshold is set to eight detected cases and there is no school closure threshold. The epidemic is simulated for 100 days. For an increase in the school size, the attack rate increases and the NSDL decreases.

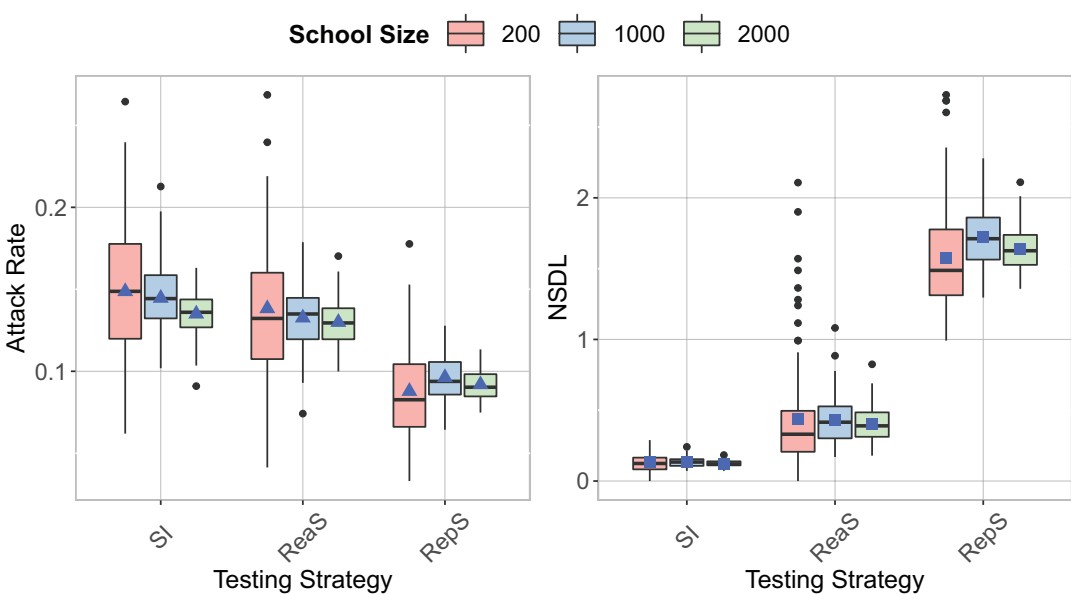

**Appendix 1—figure 18.** We show the testing strategies in the context of the Delta VoC for a seeding of 1 case per week for a school of size 200, 5 seeds per week for a school of size 1000 and 10 seeds per week for a school of size 2000. The class closure threshold is set to eight detected cases and there is no school closure threshold. The epidemic is simulated for 100 days. This experiment shows that a high stochasticity for a small school size, but similar attack rates and NSDL when the proportion of seeds over the school size is the same.

## No school and class closures

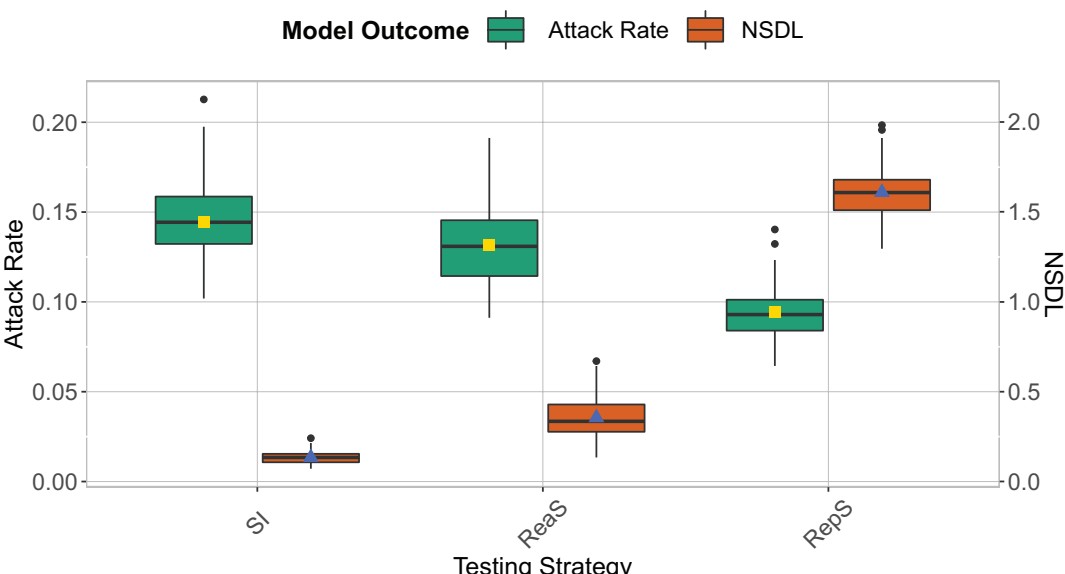

**Appendix 1—figure 19.** We show the testing strategies in the context of the Delta VoC for a moderate seeding of 5 seed per week when no class and school closures are considered. The epidemic is simulated for 100 days. This experiment shows that repetitive testing decreases the attack rate while increasing the NSDL compared to the other testing strategies.

## Omicron scenario

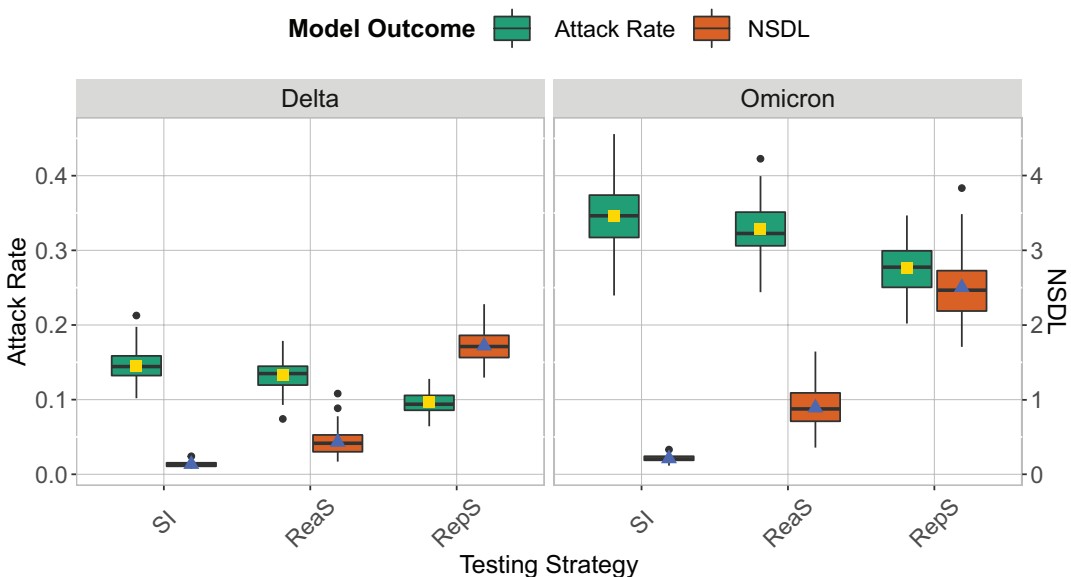

**Appendix 1—figure 20.** We show the testing strategies in the context of the Delta and Omicron variants for a moderate seeding of 5 seeds per week, when no school closure is considered and the class threshold is set to value 8. The epidemic is simulated for 100 days. Omicron is implemented by lowering the immunity proportion to 0.1 for children and 0.5 for adults, and by considering a shorter incubation period (mean 3.3 days and standard deviation of 2.2 days). This experiment shows that it is more difficult to contain the Omicron strain in a school setting. The repetitive screening strategy is shown to decrease the attack rate compared to reactive screening and symptomatic isolation while increasing the NSDL.

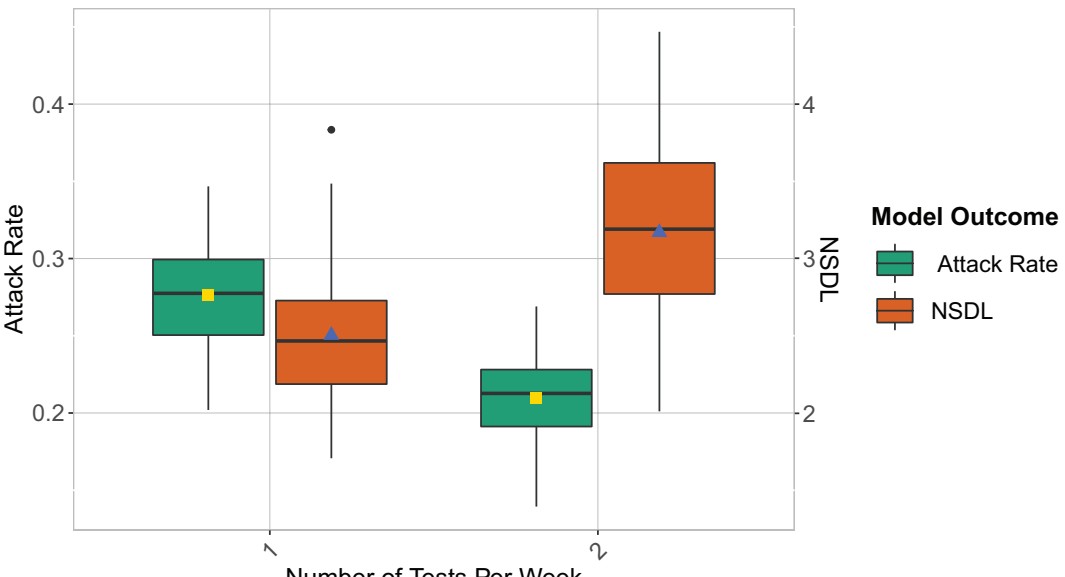

**Appendix 1—figure 21.** We show the repetitive testing strategy in the context of the Omicron VoC for a moderate seeding of 5 seeds per week, where we consider a repetitive testing strategy where the entire school population is tested either once or twice per week. We consider class closure threshold of 8 and no school closure threshold. The epidemic is simulated for 100 days. This experiment demonstrates that in the contest of Omicron, repetitive testing can further reduce the number of transmissions at school when twice testing per week is considered.

No testing scenario

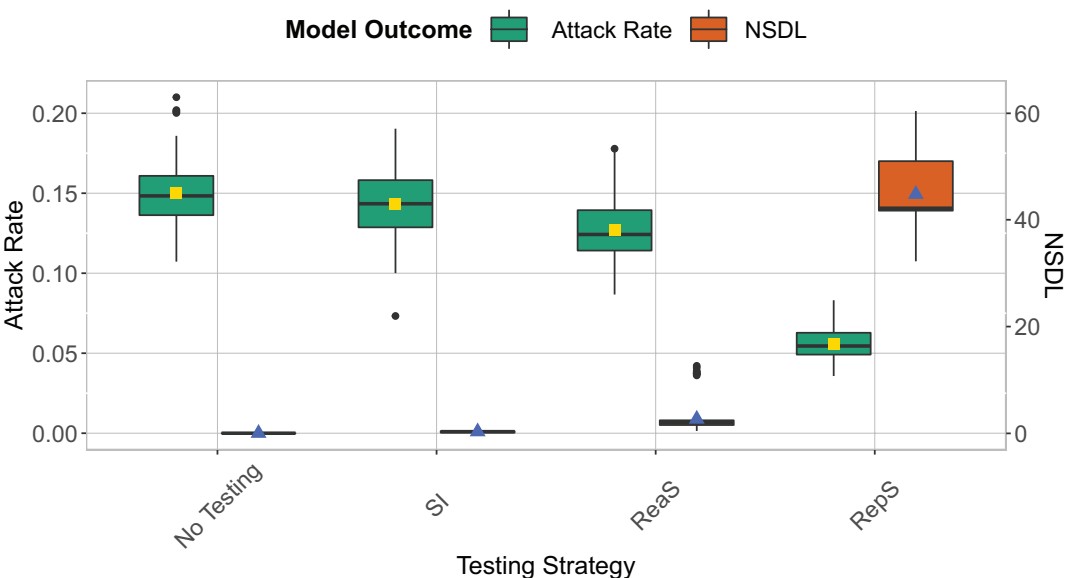

**Appendix 1—figure 22.** We show the testing strategies compared to a no-testing scenario (No Testing) in the context of the Delta variants for a moderate seeding of 5 seeds per week. when school and class closures are set, respectively, to 20 and 2 detected cases. The epidemic is simulated for 100 days. This experiment shows that symptomatic isolation and the no testing scenario have a similar performance.

## Class closure threshold: symptomatic isolation

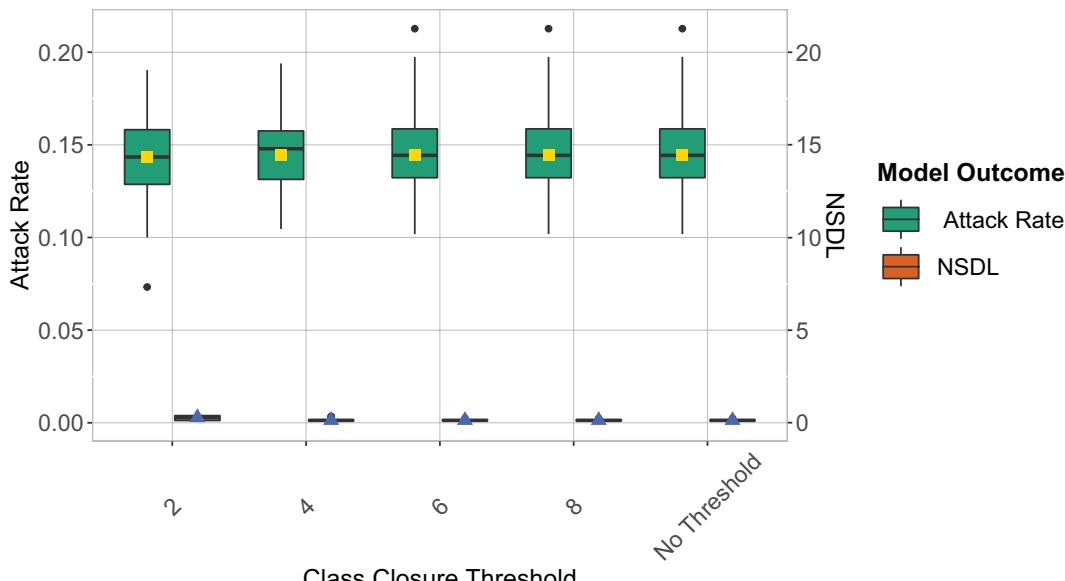

**Appendix 1—figure 23.** We show the symptomatic isolation strategy in the context of the Delta VoC for a moderate seeding of 5 seeds per week, where we consider different class closure thresholds, and no school closure threshold. The epidemic is simulated for 100 days. We note that in this experiment symptomatic isolation performs similarly for different class closure thresholds.

## Class closure threshold: reactive Screening

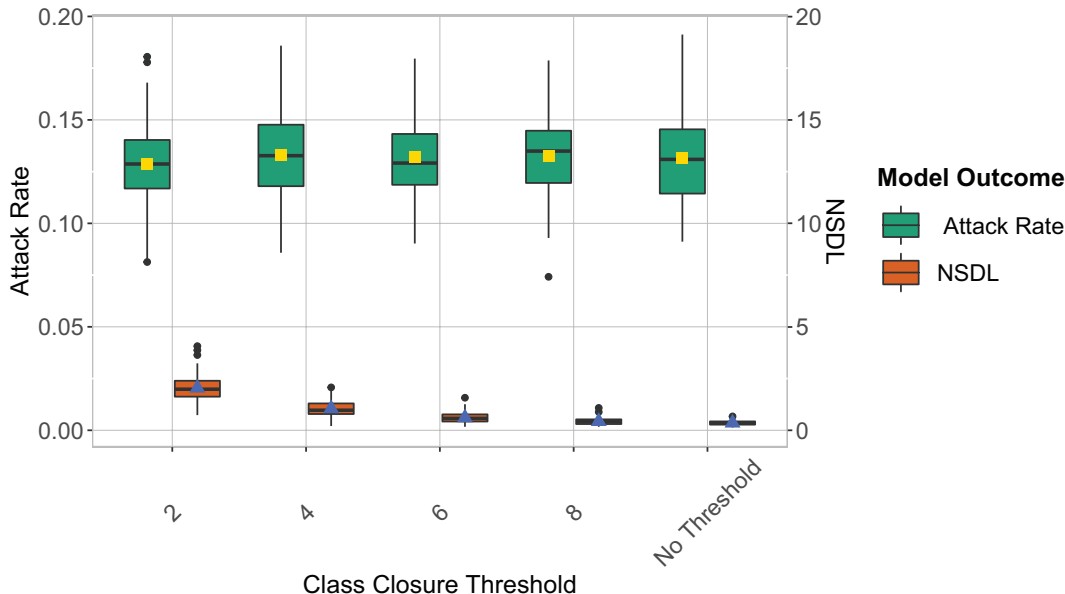

**Appendix 1—figure 24.** We show the reactive screening strategy in the context of the Delta VoC for a moderate seeding of 5 seeds per week, where we consider different class closure thresholds, and no school closure threshold. The epidemic is simulated for 100 days. We note that in this experiment symptomatic isolation performs similarly for different class closure thresholds.

