## [Editor Report]

This paper evaluates different testing strategies on the SARS-CoV-2 transmission dynamics in a primary school environment and shows that repetitive testing significantly reduces the infection attack rates in the schools. It provides insights into policy design to keep schools open as much as possible in the era of transition from COVID pandemic to endemic.

---

## [Decision Letter]

**Decision letter after peer review:**

Thank you for submitting your article "Controlling SARS-CoV-2 in schools using repetitive testing strategies" for consideration by *eLife*. Your article has been reviewed by 3 peer reviewers, one of whom is a member of our Board of Reviewing Editors, and the evaluation has been overseen by David Serwadda as the Senior Editor. The following individual involved in review of your submission has agreed to reveal their identity: Bernadette C Young (Reviewer #2).

Essential revisions:

Reviewer #1's points in brief:

1. Frequency of tests in the strategies.

2. Thresholds of class and school closure, and comparison across strategies.

3. Comparison of low, medium, and high prevalence and infection attack rates.

4. Proportion of symptomatic infections among children.

5. Parameters about the virus such as: (a) Δ vs. Omicron and (b) shorter incubation or latent period.

Reviewer #2's major points in brief:

1. Thresholds of class and school closure, and comparison across strategies.

2. Parameters about the virus such as: (a) Δ vs. Omicron and (b) reinfection.

3. Test sensitivity and turnaround time.

4. Compliance of taking tests.

5. Infection attack rates and Number of School Days Lost.

6. Frequency of tests in the Repetitive Screening strategy.

Reviewer #3's major points in brief:

1. Control baseline and infection attack rates.

2. Parameters about the virus such as: (a) Δ vs. Omicron.

3. Class or school size.

4. Consideration of severe diseases and long COVID.

5. Duration of class/school closure.

*Reviewer #1 (Recommendations for the authors):*

1. For the reactive screening strategy, it is not clear why it was assumed that the whole class would be screened only once, if one symptomatic individual was identified. It is not surprising that the reactive screening strategy was only slightly better than symptomatic isolation because every child was only screened once. Could you please explain in more detail why repeat testing was not considered when the reactive screening strategy was triggered? This strategy is still different from repetitive screening.

2. It is not clear how the thresholds of class or school closure were selected (i.e., 2 for class closure and 20 for school closure). While I understand that a higher threshold (to trigger class or school closure) would have little effect on the attack rate (Figure 2), a lower threshold should also be assessed in the sensitivity analysis, such as 2 for school closure. Lowering the closure threshold would have more substantial effects on attack rates and NSDL, and thus an "optimal" threshold could be potentially identified to minimize the attack rates or NSDL.

3. It is not clear why "continuous" weekly introduction events to the school was assumed, by seeding 1, 5, or 10 cases weekly. Since continuous introduction was assumed, it is quite intuitive that the repetitive screening strategy is the most effective, because none of the three strategies would be able to "eliminate" infections for more than a week. In a low prevalence setting such as seeding 1 case every month, is repetitive screening still more effective than reactive screening or symptomatic screening?

4. How sensitive are the results to the assumption about the proportion of symptomatic infections among children? This also matters more in a low prevalence school setting.

5. How sensitive are the results to a shorter incubation period and/or a shorter latent period? Incorporating this assumption in the sensitivity analysis would be potentially useful to guide public health practice.

Examples of a few similar studies to be included in the literature review in the introduction:

a) Chang JT, Crawford FW, Kaplan EH. Repeat SARS-CoV-2 testing models for residential college populations. Health care management science. 2021 Jun;24(2):305-18.

b) Hamer DH, White LF, Jenkins HE, Gill CJ, Landsberg HE, Klapperich C, Bulekova K, Platt J, Decarie L, Gilmore W, Pilkington M. Assessment of a covid-19 control plan on an urban university campus during a second wave of the pandemic. JAMA Network Open. 2021 Jun 1;4(6):e2116425-.

c) Paltiel AD, Schwartz JL. Assessing COVID-19 prevention strategies to permit the safe opening of residential colleges in Fall 2021. Annals of internal medicine. 2021 Nov;174(11):1563-71.

*Reviewer #2 (Recommendations for the authors):*

This study uses mathematical models to assess the likely outcome of 3 models of testing on two outcomes in a primary school like setting (where students are in a single class and teachers teach a single class).

– Number people infected by a case in school (attack rate).

– Number of school days missed.

Three strategies are compared

– Testing individuals who develop symptoms.

– Testing individuals who develop symptoms, and if positive, their class.

– Testing all individuals repetitively. The paper does not specify what this interval is, but later results suggest the starting assumption is weekly.

A strength of this approach is the ability to predict how strategies might work based on what we know about viral transmission. The models are clearly defined, and allows others to repeat the work. However as with all modelling, the study is based on a number of assumptions, some of which are widely accepted based on available evidence.

– Not all cases are detected, and children less likely to be detected than adults.

– That individuals are most infectious when they develop symptoms.

Some of these assumptions are specific to public health measures that do not necessarily apply across other settings.

– That 2 cases in a class within 14 days would lead to closing the class, and 20 cases in a rolling 14 day window would close the school.

– 10 days isolation of cases (now lower in many parts of the world).

One of the primary outcomes (days of school missed) would be very sensitive to these policies, especially class closure. This policy is an important part of the intervention, and should be highlighted in abstract and title to aid with interpreting the results. Sensitivity analysis has been performed around these assumptions, however I have concerns with the clarity and accuracy of the presentation of the results of this analysis, which I will discuss in the findings.

Additional assumptions, which have not been tested by sensitivity analysis in this paper, do give me some concern about the applicability of the model.

The viral parameters are modelled for original pandemic and Δ strains, and the model makes assumptions about high immune rates in adults, lower in children, reasonable in settings of high vaccination rates. However it also assumes reinfection does not occur, which is reasonable over the 100 days of the model but not with a newly introduced VOC or over longer time periods.

The model further assumes that testing in clinical practice has imperfect sensitivity (83%) and turn- around time of 1 day. This latter is plausible at some times but not in all settings, and often not at times of high case rates. The sensitivity analysis should include delays to turn around times.

Finally, the model assumes all invited to testing take part, which is unlikely to reflect clinical reality, particularly for asymptomatic testing in the repetitive testing strategy, and evidence is needed to support this assumption.

In the model 5 cases in a school acquired infection outside school are imported, and cases and absence over the next 100 days are computed. The model runs 100 times to determine cases and absences.

The authors show a large reduction in the attack rate is lower if screening is pursued, and greater impact with reactive screening. Reactive (ReaS) screening has a limited effect, likely due to incomplete case ascertainment (high number cases not being detected due to high number asymptomatic). Repetitive screening (RepS) leads to average 4 as the Number of School Days Lost (NSDL) in 100 days for Wuhan and over 40 /100 in Δ. This means students would miss almost half days in education.

The effect of changing these thresholds is reviewed, and notably removing school closure threshold and reducing class thresholds does reduce NSDL. However Figure 2 shows results only for RepS, with Δ, and does not show a direct comparison with other testing strategies. However for all the results presented, the upper bound of the estimated attack rate are now within the confidence interval for the AR in ReaS. The paper does not present numbers for the results, but from graphical representation the Δ AR in ReaS has a median around 13% with an observed range of around 8-17%. The Δ AR for RepS has an average 5% with the original closure rules, but median increases to around 7.5 -8-5 with the changes in closure rules and upper bound extends to about 12%, so well within the interquartile range of the RepS strategy. I therefore disagree that author's statement at line 198 that "a higher class closure threshold has little effect on the attack rate, yet it significantly reduces the NSDL " is supported by the presented data.

Twice weekly screening was more effective than weekly in the model and with permissive rules on class closure did not increase school days missed, which is an important result. Given that other published modelling has assumed twice weekly screening (eg https://assets.publishing.service.gov.uk/government/uploads/system/uploads/attachment_data/file/1013533/S1355_SPI-M-O_Consensus_Statement.pdf), it is unclear why weekly screening would be the assumed parameter.

The authors report that RepS is 'the most efficient to reduce attack rate' – and while it has the greatest impact on attack rate, the high cost of school days missed in the core model leads me to question the 'efficiency' of the strategy. Further the qualifying statement "Simulations indicate that such a testing strategy limits the number of transmission events even when no class and school closures are in place." Is not supported by the results presented, as discussed above. Likewise, I do not think their statement "keeping transmissions under control, a limited number of school days lost is computed when no thresholds are in place" is currently supported. However if the model were investigated with twice weekly testing, this may be the case.

The authors highlight that this paper demonstrates the limitation of symptomatic based testing and isolation strategies in a population likely to have low rates of symptoms and immunity (as is currently the case in many current primary school settings).

I think this is an important piece of work and will help guide public health decision making, but some further analysis is needed to see if the findings support the authors' claims, or if the impact of repetitive screening in primary school could only be achieved with very stringent school closure rules, which would be unacceptable to many.

Abstract

– Suggest editing abstract to allow reader to determine the findings (and effect size) in abstract.

– Since isolation and closure policies are important in the model and for the primary outcome, these should be included or indicated in the abstract.

Introduction

– The piece should be placed in context of other published work modelling impact of testing strategies on school attack rate and absence, for eg (not exhaustive).

– UK SPI-MO reports around mass testing, and secondary school based testing (both mass screening) and reactive

– https://www.gov.uk/government/publications/spi-m-mass-testing-of-the-whole-population-25-november-2020

https://assets.publishing.service.gov.uk/government/uploads/system/uploads/attachment_data/file/1013533/S1355_SPI-M-O_Consensus_Statement.pdf

https://assets.publishing.service.gov.uk/government/uploads/system/uploads/attachment_data/file/976324/S1146_SPI-M-O_Daily_contact_testing.pdf

Leng et al., have presented extensive modelling evidence in secondary schools

https://www.medrxiv.org/content/10.1101/2021.07.09.21260271v1.full

https://www.epicx-lab.com/uploads/9/6/9/4/9694133/ms-testing_school-rev.pdf

– Suggest remove results from introduction section Lines 52-64, introduction should not contain results.

Methods

– Number days of school lost is not defined, nor the method for calculating defined.

– Attack rate should be defined in the methods. From results I infer it is number of cases in the school?

– Testing frequency in RepS should be made clear.

Results

– Results section should outline results prior to drawing conclusions about them (eg line 131-134) makes a comparative statement about results not yet presented.

– Results should be presented quantitatively as well as graphically, so that the reader is not left trying to deduce them.

– Figure 2 should allow comparison to other models performance, as I believe that with relaxed school/class closure rules ReaS and RepS will not be strongly different.

Discussion

– Line 219 "keeping transmissions under control, a limited number of school days lost is computed when no thresholds are in place" is currently supported, because the median attack rate was close to 10% in these simulations. However if the model were investigated with twice weekly testing, this may be the case. I encourage the authors to apply twice weekly testing to the base model of RepS and see if RepS provides a clearer benefit.

– Paragraph at line 221 is unclear. Appears the authors recounting recommendations from another paper, but this may be a matter of linguistic confusion.

– Paragraph at line 242 is unclear, I do not know which part of the paper this refers to.

Supplement

– Suggest insertion of a qualifying comment on reinfection given observation of high rate of reinfection with Omicron variant.

*Reviewer #3 (Recommendations for the authors):*

In this manuscript the authors set out to investigate the effectiveness of different COVID-19 testing strategies within school settings to reduce infections, while also reducing the loss of in-person teaching time. The authors show that (reactive) testing based on symptoms is generally ineffective (due to large proportion of asymptomatic infections) compared to a repetitive testing strategy – but also that such a strategy should be utilised with appropriate thresholds for closing in-person education given the greater number of tests. Overall, the article is well written and the authors offer several sensitivity analyses showing the robustness of their core and timely result – that frequent testing can be used to reduce infections while keeping schools open. There are four main areas I think warrant consideration in the text: (a) a control baseline – do symptomatic isolation and reactive testing strategies exhibit lower attack rates than in scenarios with no testing – it is currently unclear how (in)effective these strategies are? (b) how well do the results hold up in the context of new variants e.g. Omicron? (c) how does school size impact findings? (d) while infection may be mild for most children, neither non-mild infection and/or persistent symptoms (long COVID) are built into the modeling framework – both are additional reasons for wanting to minimize the attack rate (as is reducing community transmission), while additionally minimizing disruption to education and should be discussed.

One key detail I was unable to find in the text, how long are school/class closures implemented for when triggered?

I would suggest that the authors include the length of epidemic simulations (100 days) for figures where NSDL is reported – or maybe consider reporting this result as a proportion of school days? Also, knowing that 5 infections are seeded per a week into a population of 1000 is useful for interpretation.

The authors note that the attack rate does not include seeded infections; could the authors clarify how they calculate this?

I found it unclear in the Supplemental description what the operational differences are between pupils and teachers in the model, could the authors clarify?

I thank the authors for making their simulation code available online. I would encourage the authors to also archive their code (e.g. via Zenodo/Figshare) to create a version of record with a persistent identifier (https://docs.github.com/en/repositories/archiving-a-github-repository/referencing-and-citing-content). This would create a resource that is accessible even if GitHub goes away, changes its domain name etc. For the purposes of replication I would also encourage the authors to list which version of R and its packages were used in the code.

I would suggest the authors change "his or her" to "their" in the Supplement (line: 318)

With respect to school size – I wonder if 1000 pupils is a typical size for a primary school? In my experience many primary schools are much smaller than this, is there a particular reason this number was chosen? I think school size influences the manuscript in two ways: (a) school closure threshold (just Figure 1?) (b) appropriate importation rate of infections. So I do expect the key finding of the value of repetitive testing to remain. Are other results affected? If so, I think it would be useful to communicate some of these nuances, especially with respect to those who wish to translate these research results into actionable policy.

---

## [Author Response]

Essential revisions:Reviewer #1's points in brief:1. Frequency of tests in the strategies.2. Thresholds of class and school closure, and comparison across strategies.3. Comparison of low, medium, and high prevalence and infection attack rates.4. Proportion of symptomatic infections among children.5. Parameters about the virus such as: (a) Δ vs. Omicron and (b) shorter incubation or latent period.Reviewer #2's major points in brief:1. Thresholds of class and school closure, and comparison across strategies.2. Parameters about the virus such as: (a) Δ vs. Omicron and (b) reinfection.3. Test sensitivity and turnaround time.4. Compliance of taking tests.5. Infection attack rates and Number of School Days Lost.6. Frequency of tests in the Repetitive Screening strategy.Reviewer #3's major points in brief:1. Control baseline and infection attack rates.2. Parameters about the virus such as: (a) Δ vs. Omicron.3. Class or school size.4. Consideration of severe diseases and long COVID.5. Duration of class/school closure.

All the essential revision points listed above are addressed in the present revision. For a clearer response to the reviewers comments, we report our responses directly below each comment made by the reviewers.

Reviewer #1 (Recommendations for the authors):1. For the reactive screening strategy, it is not clear why it was assumed that the whole class would be screened only once, if one symptomatic individual was identified. It is not surprising that the reactive screening strategy was only slightly better than symptomatic isolation because every child was only screened once. Could you please explain in more detail why repeat testing was not considered when the reactive screening strategy was triggered? This strategy is still different from repetitive screening.

We selected a strategy based on a single screening of the class in which a case is detected to represent a strategy that is used in practice in different European countries. In the first version of the paper, we did not consider additional screenings because the majority of cases was supposedly detected with the first screening. To get more insights on the effect of multiple screening rounds, we extended the simulation model and we ran a scenario in which classes of detected cases are screened 2 or 3 times. Results show that multiple screenings have a low impact on attack rate and number of school days lost (Figure 6). A sentence describing this experiment has been added to the Result section (lines: 174-176).

2. It is not clear how the thresholds of class or school closure were selected (i.e., 2 for class closure and 20 for school closure). While I understand that a higher threshold (to trigger class or school closure) would have little effect on the attack rate (Figure 2), a lower threshold should also be assessed in the sensitivity analysis, such as 2 for school closure. Lowering the closure threshold would have more substantial effects on attack rates and NSDL, and thus an "optimal" threshold could be potentially identified to minimize the attack rates or NSDL.

There was no commonly defined protocol in the literature, nor among different European countries that we could use to inform the threshold values. Therefore, we had to make an initial ad-hoc choice, and we assumed a threshold of 2 for class closure and 20 for school closure. Acknowledging the uncertainty around this choice, we challenged both class and school closure thresholds in a sensitivity analysis, of the initial revision. As suggested by the reviewer, we extended this sensitivity analysis on the school closure threshold by considering a wider range of thresholds. The additional results demonstrate the effect of a school closure threshold on attack rate and NSDL. This shows that a low school closure threshold induces a low attack rate in combination with a high NSDL (Figures 11, 12 and 13).

3. It is not clear why "continuous" weekly introduction events to the school was assumed, by seeding 1, 5, or 10 cases weekly. Since continuous introduction was assumed, it is quite intuitive that the repetitive screening strategy is the most effective, because none of the three strategies would be able to "eliminate" infections for more than a week. In a low prevalence setting such as seeding 1 case every month, is repetitive screening still more effective than reactive screening or symptomatic screening?

We assumed a continuous weekly introduction to describe a context of a high COVID19 prevalence. Testing strategies are expected to be more relevant when the virus circulates widely. However, following the comment of the reviewer we simulate a scenario with a single introduction per month, and results are added to the sensitivity analysis (Figure 14). In such a scenario, the attack rate is low for all the testing strategies. Nevertheless, repetitive screening still performs better in decreasing the attack rate. We added a sentence discussing this analysis in the result section (lines 228-232).

4. How sensitive are the results to the assumption about the proportion of symptomatic infections among children? This also matters more in a low prevalence school setting.

Thank you for this question. We added a simulation scenario in which the assumption about the proportion of symptomatic individuals was varied (Figure 15). Overall, both the attack rate and the NSDL increase, when the probability of children being symptomatic is higher. Reactive screening has a larger impact on the attack rate compared to symptomatic isolation when the probability of symptomatic infection is larger. However, even when the probability of symptomatic infection increases, repetitive screening remains the strategy that most reduces the number of infections in all of the tested scenarios. We added a sentence discussing this analysis in the result section (lines 228-232).

5. How sensitive are the results to a shorter incubation period and/or a shorter latent period? Incorporating this assumption in the sensitivity analysis would be potentially useful to guide public health practice.

We extended the simulation model to enable a shorter incubation period, that we set in line with observation for the BA.1 Omicron VoC (Backer et al., 2022), and we ran a simulation comparing such incubation period with the baseline (Figure 16). Results indicate that testing strategies are less effective in decreasing the attack rate when the incubation period is shorter. In this scenario, repetitive screening remains the strategy with most impact on the number of infections. We added this scenario to the sensitivity analysis (Figure 16), and we report the findings in the result section (lines: 241-242).

Examples of a few similar studies to be included in the literature review in the introduction:a) Chang JT, Crawford FW, Kaplan EH. Repeat SARS-CoV-2 testing models for residential college populations. Health care management science. 2021 Jun;24(2):305-18.b) Hamer DH, White LF, Jenkins HE, Gill CJ, Landsberg HE, Klapperich C, Bulekova K, Platt J, Decarie L, Gilmore W, Pilkington M. Assessment of a covid-19 control plan on an urban university campus during a second wave of the pandemic. JAMA Network Open. 2021 Jun 1;4(6):e2116425-.c) Paltiel AD, Schwartz JL. Assessing COVID-19 prevention strategies to permit the safe opening of residential colleges in Fall 2021. Annals of internal medicine. 2021 Nov;174(11):1563-71.

We thank the reviewer for pointing out studies on testing strategies in school settings. We acknowledge them in the introduction to highlight the current state of the art (lines 46-49).

Reviewer #2 (Recommendations for the authors):This study uses mathematical models to assess the likely outcome of 3 models of testing on two outcomes in a primary school like setting (where students are in a single class and teachers teach a single class).– Number people infected by a case in school (attack rate).– Number of school days missed.Three strategies are compared– Testing individuals who develop symptoms.– Testing individuals who develop symptoms, and if positive, their class.– Testing all individuals repetitively. The paper does not specify what this interval is, but later results suggest the starting assumption is weekly.A strength of this approach is the ability to predict how strategies might work based on what we know about viral transmission. The models are clearly defined, and allows others to repeat the work. However as with all modelling, the study is based on a number of assumptions, some of which are widely accepted based on available evidence.– Not all cases are detected, and children less likely to be detected than adults.– That individuals are most infectious when they develop symptoms.Some of these assumptions are specific to public health measures that do not necessarily apply across other settings.– That 2 cases in a class within 14 days would lead to closing the class, and 20 cases in a rolling 14 day window would close the school.– 10 days isolation of cases (now lower in many parts of the world).One of the primary outcomes (days of school missed) would be very sensitive to these policies, especially class closure. This policy is an important part of the intervention, and should be highlighted in abstract and title to aid with interpreting the results. Sensitivity analysis has been performed around these assumptions, however I have concerns with the clarity and accuracy of the presentation of the results of this analysis, which I will discuss in the findings.

We revised the manuscript by highlighting the role of the number of school days lost, as suggested by the reviewer. We included the findings and main assumptions in the abstract, and we added several sensitivity analyses to better investigate how this quantity is affected by the testing strategies in different scenarios. We also improved the presentation of the results, increasing the clarity of the investigation and better supporting the statements made.

Additional assumptions, which have not been tested by sensitivity analysis in this paper, do give me some concern about the applicability of the model.

The goal of the developed simulation model is to obtain general insights on specific aspects of the transmission dynamics, and we agree with the reviewer that to do so the assumptions made need to be challenged in a sensitivity analysis. Following the comments of the reviewer we extended our investigation, by further challenging the assumptions made.

The viral parameters are modelled for original pandemic and Δ strains, and the model makes assumptions about high immune rates in adults, lower in children, reasonable in settings of high vaccination rates. However it also assumes reinfection does not occur, which is reasonable over the 100 days of the model but not with a newly introduced VOC or over longer time periods.

We thank the reviewer for the comment. To investigate the effect of a possible immune evasive VoC we challenged the assumption on immunity rates (Figures 8 and 9). Results show that lower immunity corresponds to higher attack rates and higher NSDL. We included these results in the result section (lines 228-232). It would be interesting to include reinfections and simulate testing strategies in longer time periods, but this requires additional model developments. Furthermore, as pointed out by the reviewer, it is reasonable not to consider reinfection in our simulation scenarios, since we look at a time interval of 100 days.

The model further assumes that testing in clinical practice has imperfect sensitivity (83%) and turn- around time of 1 day. This latter is plausible at some times but not in all settings, and often not at times of high case rates. The sensitivity analysis should include delays to turn around times.

We added a sensitivity analysis investigating the turn around time of 1,2,3 days. We observed an increase in the attack rate for the reactive screening and the repetitive testing strategies with an increase in the turn around time, while a similar trend is observed for the symptomatic isolation strategy (Figure 17). We report these findings in the result section (lines 236-241).

Finally, the model assumes all invited to testing take part, which is unlikely to reflect clinical reality, particularly for asymptomatic testing in the repetitive testing strategy, and evidence is needed to support this assumption.

We agree that compliance to testing is an important assumption, especially for asymptomatic testing or in low-prevalence settings. Therefore, we extended the simulation model to include compliance to testing and we ran a sensitivity analysis in which we vary the compliance to testing for the repetitive screening strategy. Overall, for a decrease in compliance the attack rate increases and the NSDL decreases (Figure 18). Interestingly, in our experiment we observe a similar attack rate between a 60% and 100% compliance. We added a sentence about compliance in the result section (lines 252-256). In addition, we rephrased the paragraph about compliance in the Discussion section according to the result obtained in the sensitivity analysis (lines 310-317).

In the model 5 cases in a school acquired infection outside school are imported, and cases and absence over the next 100 days are computed. The model runs 100 times to determine cases and absences.The authors show a large reduction in the attack rate is lower if screening is pursued, and greater impact with reactive screening. Reactive (ReaS) screening has a limited effect, likely due to incomplete case ascertainment (high number cases not being detected due to high number asymptomatic). Repetitive screening (RepS) leads to average 4 as the Number of School Days Lost (NSDL) in 100 days for Wuhan and over 40 /100 in Δ. This means students would miss almost half days in education.The effect of changing these thresholds is reviewed, and notably removing chooll closure threshold and reducing class thresholds does reduce NSDL. However Figure 2 shows results only for RepS, with Δ, and does not show a direct comparison with other testing strategies. However for all the results presented, the upper bound of the estimated attack rate are now within the confidence interval for the AR in ReaS. The paper does not present numbers for the results, but from graphical representation the Δ AR in ReaS has a median around 13% with an observed range of around 8-17%. The Δ AR for RepS has an average 5% with the original closure rules, but median increases to around 7.5 -8-5 with the changes in closure rules and upper bound extends to about 12%, so well within the interquartile range of the RepS strategy. I therefore disagree that author’s statement at line 198 that “a higher class closure threshold has little effect on the attack rate, yet it significantly reduces the NSDL “ is supported by the presented data.

We apologise for the confusion. To facilitate the comparison among testing strategies when different class closure thresholds are considered, we added Figures 25 and 26 presenting the results of a sensitivity analysis in which symptomatic isolation and reactive screening are considered.

Furthermore, we report both the median and the 95% quantile interval in the result section (Table 2), to allow for a clearer interpretation. By specifying these values, we support the statement that the NSDL significantly reduces.

We changed the sentence:

“This experiment shows that a higher class closure threshold has little effect on the attack rate, yet it significantly reduces the NSDL” into

“This experiment shows that when repetitive testing is in place, a higher class closure threshold has little effect on the attack rate, yet it significantly reduces the mean NSDL (Table 2). (line 259)”.

Twice weekly screening was more effective than weekly in the model and with permissive rules on class closure did not increase school days missed, which is an important result. Given that other published modelling has assumed twice weekly screening (eg https://assets.publishing.service.gov.uk/government/uploads/system/uploads/attachment_data/file/1013533/S1355_SPI-M-O_Consensus_Statement.pdf), it is unclear why weekly screening would be the assumed parameter.

We assumed a weekly screening as the baseline scenario because a strategy based on a single test can be more easily applied at national level when a large number of tests need to be quickly analyzed. We added a sentence in the manuscript to clarify our choice (lines 96-100). Furthermore, by keeping this approach we stress even more that even a single test per week is sufficient for a repetitive testing strategy to perform better than symptomatic isolation and reactive screening in reducing transmission. However, we acknowledge that twice weekly testing is suitable to further reduce the attack rate and we show the effect of such a strategy in two scenarios in which we consider, respectively, the Δ VoC (Figure 4,5), and the Omicron VoC (Figure 23). For a twice weekly testing strategy in the setting of the Δ Voc, we noticed a decrease in the NSDL when the assumption on school and class closure are relaxed compared to a single repetitive testing strategy (Figure 5). However, the opposite trend is observed when school and class thresholds are present (Figure 4), or when the Omicron VoC is considered (Figure 23). We describe this in the result section (lines 242-252).

The authors report that RepS is 'the most efficient to reduce attack rate' – and while it has the greatest impact on attack rate, the high cost of school days missed in the core model leads me to question the ‘efficiency’ of the strategy. Further the qualifying statement “Simulations indicate that such a testing strategy limits the number of transmission events even when no class and school closures are in place.” Is not supported by the results presented, as discussed above.

We agree with the reviewer and included a sensitivity analysis in which the three testing strategies are considered when no school or class closure thresholds are in place (Figure 21). Results show that repetitive testing is the most successful intervention under study in reducing the attack rate.

Likewise, I do not think their statement “keeping transmissions under control, a limited number of school days lost is computed when no thresholds are in place” is currently supported. However if the model were investigated with twice weekly testing, this may be the case.

We included the sensitivity analysis in which no school and class closures are considered (Figure 21) to support the claim. Furthermore we added a scenario in which school threshold is varied for the reactive screening and symptomatic isolation strategies (Figures 11,12 and 13), showing that such strategies can reach a similar control to repetitive screening only with a low school closure thresholds, leading to a high NSDL. We also included two additional scenarios in which twice weekly testing is considered (Figures 4 and 22). We noticed that while in one case the NSDL are further reduced compared to testing on a weekly basis (Figure 5), in the other scenario we observe an increase in the NSDL (Figure 4, and 22).

The authors highlight that this paper demonstrates the limitation of symptomatic based testing and isolation strategies in a population likely to have low rates of symptoms and immunity (as is currently the case in many current primary school settings).I think this is an important piece of work and will help guide public health decision making, but some further analysis is needed to see if the findings support the authors’ claims, or if the impact of repetitive screening in primary school could only be achieved with very stringent school closure rules, which would be unacceptable to many.

We thank the reviewer for the appreciation of our work, and for the comments made which helped to improve the manuscript. We believe that the revised manuscript addresses the comments made, further clarifying and supporting the claims made.

Abstract– Suggest editing abstract to allow reader to determine the findings (and effect size) in abstract.– Since isolation and closure policies are important in the model and for the primary outcome, these should be included or indicated in the abstract.

We changed the following sentence of the abstract:

“Through this analysis, we demonstrate that repetitive testing strategies can significantly reduce the attack rate in schools, contrary to a reactive screening approach. Furthermore, we investigate the impact of these testing strategies on the average number of school days lost per child.” Into:

“Through this analysis, we demonstrate that repetitive testing strategies can significantly reduce the attack rate in schools, contrary to a reactive screening or a symptomatic isolation approach. However, when a repetitive testing strategy is in place, more cases will be detected and class and school closures are more easily triggered, leading to a high number of school days lost per child. While maintaining the epidemic under control with a repetitive testing strategy, absenteeism can be reduced by relaxing class and school closure thresholds.” (lines: 14-22)

Introduction– The piece should be placed in context of other published work modelling impact of testing strategies on school attack rate and absence, for eg (not exhaustive).– UK SPI-MO reports around mass testing, and secondary school based testing (both mass screening) and reactive– https://www.gov.uk/government/publications/spi-m-mass-testing-of-the-whole-population-25-november-2020https://assets.publishing.service.gov.uk/government/uploads/system/uploads/attachment_data/file/1013533/S1355_SPI-M-O_Consensus_Statement.pdfhttps://assets.publishing.service.gov.uk/government/uploads/system/uploads/attachment_data/file/976324/S1146_SPI-M-O_Daily_contact_testing.pdfLeng et al., have presented extensive modelling evidence in secondary schoolshttps://www.medrxiv.org/content/10.1101/2021.07.09.21260271v1.fullhttps://www.epicx-lab.com/uploads/9/6/9/4/9694133/ms-testing_school-rev.pdf– Suggest remove results from introduction section Lines 52-64, introduction should not contain results.

The introduction has been adapted including the suggested investigations concerning testing in schools, and removing results from the introduction (lines 46-49).

Methods– Number days of school lost is not defined, nor the method for calculating defined.– Attack rate should be defined in the methods. From results I infer it is number of cases in the school?– Testing frequency in RepS should be made clear.

We added the definition of both the number of school days lost and the attack rate as follows:

"For each simulated outbreak we compute two summary measures that account, re-

spectively, for the number of transmissions at school and absenteeism. The former is defined as the total number of cases (minus the index cases) divided by the number of pupils in the school, and we refer to this quantity as the attack rate. The latter is defined as the total number of school days lost divided by the school size, and we refer to this quantity as number of school days lost (NSDL)." (lines 131-137)

In addition we specified that repetitive testing is considered to have a frequency of once per week in the baseline scenario. (lines 85-87)

Results– Results section should outline results prior to drawing conclusions about them (eg line 131-134) makes a comparative statement about results not yet presented.– Results should be presented quantitatively as well as graphically, so that the reader is not left trying to deduce them.– Figure 2 should allow comparison to other models performance, as I believe that with relaxed school/class closure rules ReaS and RepS will not be strongly different.

We restructured the Result section according to the reviewer’s comments, and we added the median estimate together with the 95% quantile interval for a clearer presentation of the results.

We also added a scenario in the sensitivity analysis simulating the effect of class closure thresholds for the symptomatic isolation (Figure 25) and the reactive screening (Figure 26), which can be compared to Figure 2. We observed that the summary measures computed for the repetitive screening and reactive screening are different. In particular, repetitive screening performs better in reducing the attack rate when class thresholds vary, compared to reactive screening.

Discussion– Line 219 "keeping transmissions under control, a limited number of school days lost is computed when no thresholds are in place" is currently supported, because the median attack rate was close to 10% in these simulations. However if the model were investigated with twice weekly testing, this may be the case. I encourage the authors to apply twice weekly testing to the base model of RepS and see if RepS provides a clearer benefit.

We added the requested simulation to the sensitivity analysis (Figure 4). We noticed that testing twice per week further reduces the attack rate also in this case, compared to testing once per week. However, twice weekly testing increases the NSDL (Figure 4). In addition, we also investigated the use of twice weekly for the Omicron VoC (Figure 23.)

– Paragraph at line 221 is unclear. Appears the authors recounting recommendations from another paper, but this may be a matter of linguistic confusion.

We restructured the paragraph to present more clearly the point we wanted to make.

(lines 284-290)

– Paragraph at line 242 is unclear, I do not know which part of the paper this refers to.

We apologize for the confusion, and we decided to remove this paragraph, in favour of a more clear and structured Discussion section.

Supplement– Suggest insertion of a qualifying comment on reinfection given observation of high rate of reinfection with Omicron variant.

We added sensitivity analyses in which we consider the Omicron VoC and we varied the proportion of immune individuals in the sensitivity analysis.

Reviewer #3 (Recommendations for the authors):In this manuscript the authors set out to investigate the effectiveness of different COVID-19 testing strategies within school settings to reduce infections, while also reducing the loss of in-person teaching time. The authors show that (reactive) testing based on symptoms is generally ineffective (due to large proportion of asymptomatic infections) compared to a repetitive testing strategy – but also that such a strategy should be utilised with appropriate thresholds for closing in-person education given the greater number of tests. Overall, the article is well written and the authors offer several sensitivity analyses showing the robustness of their core and timely result – that frequent testing can be used to reduce infections while keeping schools open. There are four main areas I think warrant consideration in the text:a) a control baseline – do symptomatic isolation and reactive testing strategies exhibit lower attack rates than in scenarios with no testing – it is currently unclear how (in)effective these strategies are?

We included the sensitivity analysis in which we compare the testing strategies with a baseline scenario, in which individuals are not tested. Results show that symptomatic isolation performs similarly well to a no testing strategy, while reactive screening performs slightly better (Figure 24).

b) how well do the results hold up in the context of new variants e.g. Omicron?

We extended the sensitivity analysis by including a scenario which represents the Omicron variant. Omicron has been simulated by reducing the immunity for children and adults to, respectively, 0.1 and 0.5, and by including a shorter incubation period (Mean 3.3 days, sd 2.2 days). Results show that all testing strategies are less effective against the Omicron variant (Figure 22). Twice weekly testing improves the performance of repetitive testing (Figure 23). We included these results in the result section (lines 212-215).

c) how does school size impact findings?

We investigated simulation scenarios to further investigate the impact of the school size (Figures 19 and 20). Results show that the size can affect the attack rate when the same number of seeds are added among schools of different sizes. Precisely, to a lower school size, we observe a higher attack rate (Figure 19). This is possibly caused by the number of seeds, which accounts for a different proportion of the school population when the school size varies. We noticed a similar effect in Figure 3, where we tested the impact of the seeding number. To get more insights into this “seeding effect”, we simulated a scenario in which the number of seeds are weighted according to the population size. Results show that results are similar for different school sizes when we considered the weighted number of seeds (Figure 20). We describe this in the result section indicating the effect of the school size (lines 223-227).

d) while infection may be mild for most children, neither non-mild infection and/or persistent symptoms (long COVID) are built into the modeling framework – both are additional reasons for wanting to minimize the attack rate (as is reducing community transmission), while additionally minimizing disruption to education and should be discussed.

We thank the reviewer for the point made and we mention this in the discussion (lines 264-267).

One key detail I was unable to find in the text, how long are school/class closures implemented for when triggered?

We thank the reviewer for pointing this out. School/class closure, as well as isolation, are implemented for 10 days, according to the duration of viral clearance (Chang et al., 2020) and in line with the isolation policy in place in Belgium in 2021. We specified the length of the school/class closure in the methods section. (lines 91-95).

I would suggest that the authors include the length of epidemic simulations (100 days) for figures where NSDL is reported – or maybe consider reporting this result as a proportion of school days? Also, knowing that 5 infections are seeded per a week into a population of 1000 is useful for interpretation.

We added the following sentence in each caption: "The epidemic is simulated for 100 days". The number of seeds is also reported in each caption.

The authors note that the attack rate does not include seeded infections; could the authors clarify how they calculate this?

The attack rate is calculated as the total number of cases excluding the number of seeds divided by the number of pupils in the school. We explicitly add this definition in the text (lines 133-135).

I found it unclear in the Supplemental description what the operational differences are between pupils and teachers in the model, could the authors clarify?

We clarified that teachers are assumed to be linked to a single class, while pupils are linked with both children of the same class and children of other classes (lines 391-395).

I thank the authors for making their simulation code available online. I would encourage the authors to also archive their code (e.g. via Zenodo/Figshare) to create a version of record with a persistent identifier (https://docs.github.com/en/repositories/archiving-a-github-repository/referencing-and-citing-content). This would create a resource that is accessible even if GitHub goes away, changes its domain name etc. For the purposes of replication I would also encourage the authors to list which version of R and its packages were used in the code.

As suggested, we uploaded the code in a Zenodo repository (DOI:10.5281/zenodo.6488473).

We also listed the R version used, both in the repository and in the manuscript (lines 314-316.). The repository will be made freely available after acceptance of the manuscript by the journal.

I would suggest the authors change "his or her" to "their" in the Supplement (line: 318)

We made the suggested change (line 369).

With respect to school size – I wonder if 1000 pupils is a typical size for a primary school? In my experience many primary schools are much smaller than this, is there a particular reason this number was chosen? I think school size influences the manuscript in two ways: (a) school closure threshold (just Figure 1?) (b) appropriate importation rate of infections. So I do expect the key finding of the value of repetitive testing to remain. Are other results affected? If so, I think it would be useful to communicate some of these nuances, especially with respect to those who wish to translate these research results into actionable policy.

School size depends on national or regional school system, and on whether these are in metropolitan or rural areas. We assumed a size of 1000, but we challenged this assumption in the sensitivity analysis, following the suggestion made. By investigating the point raised by the reviewer we noticed that the size can affect the attack rate when the same number of seeds are added among schools with different sizes. As discussed in the reply to comment(c), this is because of the different proportion of seeds over the population, and when weighted seeds are considered results are in line among schools with different sizes. Similarly, the school size could affect the number of school closures, when thresholds are kept the same among school with different sizes. However, repetitive testing is always expected to be the strategy that most reduces the number of transmissions at school.